# Normative evidence accumulation in unpredictable environments

**Christopher M Glaze[1,2]\*, Joseph W Kable[2], Joshua I Gold[1]**

[1]Department of Neuroscience, University of Pennsylvania, Philadelphia, United States; [2]Department of Psychology, University of Pennsylvania, Philadelphia, United States

**Abstract** In our dynamic world, decisions about noisy stimuli can require temporal accumulation of evidence to identify steady signals, differentiation to detect unpredictable changes in those signals, or both. Normative models can account for learning in these environments but have not yet been applied to faster decision processes. We present a novel, normative formulation of adaptive learning models that forms decisions by acting as a leaky accumulator with non-absorbing bounds. These dynamics, derived for both discrete and continuous cases, depend on the expected rate of change of the statistics of the evidence and balance signal identification and change detection. We found that, for two different tasks, human subjects learned these expectations, albeit imperfectly, then used them to make decisions in accordance with the normative model. The results represent a unified, empirically supported account of decision-making in unpredictable environments that provides new insights into the expectation-driven dynamics of the underlying neural signals.

## Introduction

Even the simplest perceptual judgments, like detecting the presence of a dim light, take time for the brain to process (*Luce, 1986*). Some of this time reflects sensory and motor processing, but a considerable fraction is dedicated to the decision process that converts the incoming sensory information into a categorical judgment that guides behavior (*Sternberg, 2001*). Under certain conditions, this temporally unfolding process serves a normative purpose: improving the accuracy of the decision by reducing uncertainty about the source or identity of noisy inputs. The sequential probability ratio test (SPRT), drift-diffusion model, and related sequential-sampling models are forms of 'belief-updating' rules for this normative process, based on perfect integration over time of the logarithm of the likelihood ratio (*LLR*) associated with each data point (*Barnard, 1946*; *Wald, 1947*; *Good, 1979*; *Link, 1992*; *Gold and Shadlen, 2001*; *Smith and Ratcliff, 2004*; *Bogacz et al., 2006*). These models have been useful for studying neural mechanisms of decision-making (*Gold and Shadlen, 2007*) but are normative for only a restricted set of conditions in which: (1) the ideal starting time-point for accumulation is known (e.g., given by the onset of an experimental trial); and (2) the statistics of the incoming information are perfectly stable throughout the entire sequence, with no change in the underlying signal and all noise coming from the same probability distribution.

Perfect integration can be particularly problematic for tasks that require the detection of signal changes (*Clifford and Ibbotson, 2002*). When there is certainty about when the change might occur, integrated signals from before vs after that time can be compared to detect the change (*Green and Swets, 1966*; *Macmillan and Creelman, 2004*). However, when there is temporal uncertainty about the change, integrating evidence at the wrong time might miss the signal or add unnecessary noise, resulting in a loss of sensitivity to the change (*Lasley and Cohn, 1981*). Several possible solutions to this problem have been proposed, including using a leaky integrator, taking a time derivative of the evidence to identify changes, or using knowledge of the spatial and temporal structure of the stimulus to guide a more directed search for the evidence

\*For correspondence: cglaze@sas.upenn.edu

**Competing interests:** The authors declare that no competing interests exist.

**Reviewing editor**: Timothy Behrens, Oxford University, United Kingdom

**eLife digest** Organisms gather information from their surroundings to make decisions. Traditionally, neuroscientists have investigated decision-making by first asking what would be optimal for the animal, and then seeing whether and how the brain implements the optimal process. This approach has assumed that the environment consists of noisy, but stable, signals that the brain must decipher by accumulating information over time and 'averaging out' the noise.

Previous research had suggested that most animals can accumulate information. However, these studies also showed that animals, including humans, often fall short of the optimal solution by being overly sensitive to noise and failing to completely average it out. Of course, in real life, the signals themselves can change abruptly and unpredictably, challenging us to distinguish noise from changes in the underlying signals. If a moving target suddenly jolts to the right, is that change part of the normal jitter that should be ignored, or does it predict where the target will be next? How do we know when to keep old information that is still relevant to the decision, and when to discard the old information because a change might have occurred that renders it irrelevant?

Glaze et al. have addressed this question by building optimal change detection into the traditional 'information-accumulation' framework. The model suggests that what researchers previously thought was an over-sensitivity to noise might actually be optimal for the real-life challenge of detecting change. In two different tasks, Glaze et al. tested human volunteers to see if they could make decisions in ways predicted by the model. One task involved the volunteers making decisions about which one of two possible sources of noisy signals generated a given piece of information, with the correct answer changing unpredictably every 1–20 trials. The other task involved looking at a crowd of moving dots, which jolted and wobbled as they changed direction, and the volunteers had to decide which direction the dots were moving at the end of each trial.

Both experiments showed that the volunteers were remarkably good at making decisions in the ways predicted by the new model, and incorporated learned expectations about the rate of change in underlying signals. The results suggest that humans, and potentially other organisms, are capable of detecting changes in the optimal ways suggested by the decision-making model. The study also makes predictions about what kinds of neural patterns neuroscientists might find when measuring brain activity while organisms do similar tasks.

(*Henning et al., 1975*; *Nachmias and Rogowitz, 1983*; *Smith, 1995*, *1998*; *Verghese et al., 1999*; *Schrater et al., 2000*). However, none of these solutions provide more general insights into how to balance the operations used to identify both steady, noisy, signals and unpredictable changes in those signals.

Here we present a normative model of decisions between two alternatives that provides such an account. In a variety of learning and other tasks, the tradeoff between signal identification and change detection has been related to inference algorithms in hidden Markov models and other Bayesian algorithms. These algorithms estimate statistical parameters in the presence of abrupt and unpredictable change-points in the otherwise stable statistics of a data-generating process (*Zakai, 1965*; *Liptser and Shiryaev, 1977*; *Rabiner, 1989*; *Yu and Dayan, 2005*; *Adams and MacKay, 2007*; *Behrens et al., 2007*; *Fearnhead and Liu, 2007*; *Wilson et al., 2010*; *McGuire et al., 2014*; *Sato and Kording, 2014*). Here we express these algorithms in a novel form that, unlike previous change-point models, is based on the *LLR* and thus can be compared directly to standard decision models based on evidence accumulation (*Gold and Shadlen, 2001*; *Usher and McClelland, 2001*; *Smith and Ratcliff, 2004*; *Bogacz et al., 2006*). The form thus yields quantitative predictions of both choice behavior and the underlying neural signals for decisions about unstable, noisy stimuli (*Gold and Shadlen, 2007*). A key feature of the model is that the expected amount of instability in the environment governs the temporal dynamics of the decision process. When perfect stability is expected, evidence is accumulated perfectly. Otherwise, evidence is accumulated with a leak (*Usher and McClelland, 2001*) to a non-absorbing boundary that expedites the identification of unexpected changes that should re-start the accumulation process, where both the leak and the boundary depend on the level of expected instability in the environment. These expectation-dependent dynamics represent a novel view of leaky, saturating, or otherwise imperfect evidence accumulation, which here may be understood as facilitating, rather than

hindering, statistical inference. We show that human decision-makers can use these dynamics to solve two different tasks on different timescales (tens of seconds vs hundreds of milliseconds) that each requires information accumulation in the presence of unpredictable change-points occurring at different rates.

## Results

### Model

Consider a decision about which of two alternatives is the present source of a sequence of noisy data arriving over time. We derived a belief-update rule for these kinds of decisions based on Bayesian principles that have typically been used to understand learning processes in dynamic environments on relatively slow timescales (*Figure 1A*) (*Yu and Dayan, 2005*; *Adams and MacKay, 2007*; *Behrens et al., 2007*; *Fearnhead and Liu, 2007*; *Wilson et al., 2010*; *McGuire et al., 2014*; *Sato and Kording, 2014*). This rule both accounts for environmental instability and relates directly to models of perfect, leaky, and bounded accumulation that have been used to understand decision processes in stable environments (*Link, 1992*; *Gold and Shadlen, 2001*; *Usher and McClelland, 2001*; *Smith and Ratcliff, 2004*; *Bogacz et al., 2006*). We define belief as the logarithm of the posterior odds of the alternative sources of information ($L$) given all information collected until a given time point. The sign of $L$ indicates which source is currently believed to be generating the information, and the magnitude of $L$ indicates how certain that belief is. The update rule is optimal when there is a fixed probability that the source could switch to the alternative at any time (i.e., according to a Bernoulli process). Specifically,

$$L_n = \psi(L_{n-1}, H) + LLR_n,\qquad(1)$$

where $L_n$ is the belief at time step $n$, $LLR_n$ is the sensory evidence (the log likelihood ratio) at step $n$, $H$ (the 'hazard rate') is the expected probability at each time step that the source will switch from one alternative to the other, and $\psi$ is the time-varying prior expectation (the logarithm of the prior odds) about the source before observing the new evidence:

$$\psi(L_{n-1}, H) = L_{n-1} + \log\left[\frac{1-H}{H} + \exp(-L_{n-1})\right] - \log\left[\frac{1-H}{H} + \exp(L_{n-1})\right].\qquad(2)$$

The prior expectation $\psi$ is the key feature of the model, balancing integration to identify steady signals and differentiation to detect changes by dynamically filtering sensory information in a way that depends on both $L$ and $H$ (*Figures 1, 2*). For the special case of $H = 0$ (perfect stability), the two rightmost terms in *Equation 2* cancel. In this case, the update *Equation 1* reduces to perfect accumulation as in random-walk and related decision models used to identify steady, but noisy, signals (*Figure 1D*) (*Smith and Ratcliff, 2004*; *Bogacz et al., 2006*). In contrast, when $H$ is high and changes are expected, accumulation over time is severely limited to facilitate change detection (*Figures 1F, 2G*). For intermediate values of $H$, these operations trade-off to emphasize change detection at the expense of steady signal identification (for higher $H$) or vice versa (for lower $H$; *Figures 1E, 2G*). Finally, in the special case of $H = 0.5$, the history of evidence is irrelevant at all times and all three terms in *Equation 2* cancel, so $\psi = 0$ and $L_n = LLR_n$.

To gain further insight into the dynamics of the model and how it controls this trade-off, we made approximations of the nonlinearity in *Equation 2*:

$$\psi(L_{n-1}, H) \approx (1 - K_n) \times L_{n-1} + \theta_n,\qquad(3)$$

$$\approx (1 - 2H) \times L_{n-1} \text{ when } L_{n-1} \approx 0,\qquad(3a)$$

$$\approx \log[(1-H)/H]\text{when } L_{n-1} \gg 0,\qquad(3b)$$

$$\approx -\log[(1-H)/H] \text{ when } L_{n-1} \ll 0.\qquad(3c)$$

Here $K_n$ governs the leakiness of the accumulation process, and $\theta_n$, governs a bias. Both parameters are adaptive, depending on both $H$ and $L_n$, with dynamics that jointly establish a boundary on the prior and thus limit subsequent belief strength. The dynamics include two regimes, as follows.

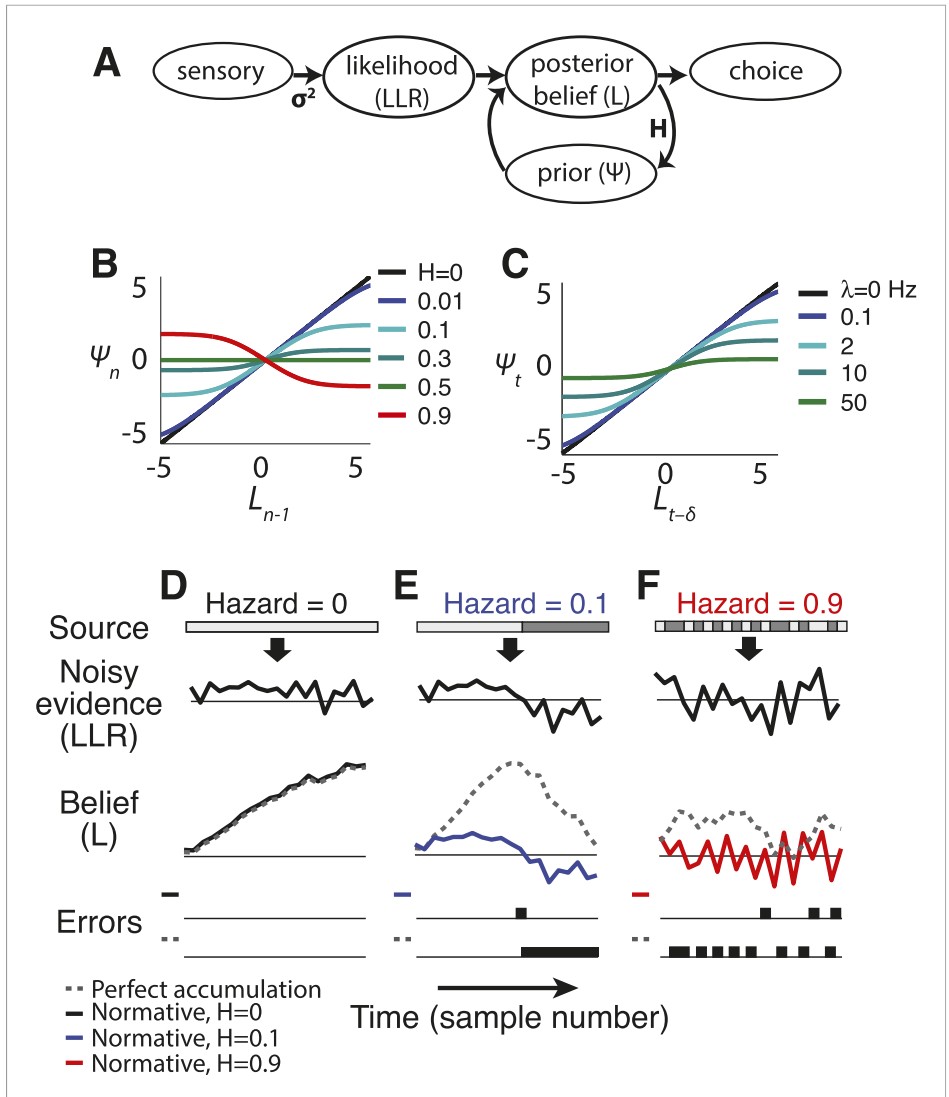

**Figure 1**. Normative model. (**A**) Illustration of the belief-updating process. (**B**) Discrete-time log-prior odds at a given moment as a function of the belief at the prior moment, plotted as *Equation 2* for different values of *H*. (**C**) Continuous-time version of the model, with log-prior odds plotted as a function of belief, computed by numerically integrating *Equation 4* with dx(t) = 0 over a 16 ms interval. Here expected instability ($\lambda$) has units of number of changes per s. Thus, $\lambda \rightarrow \infty$ is analogous to discrete-time $H \rightarrow 0.5$. (**D**–**F**) Examples of how the normative model (solid lines) and perfect accumulation (dashed gray lines) process a time-dependent stimulus (light vs dark grey for the two alternatives, shown at the top) for different hazard rates (*H*).
The following figure supplement is available for figure 1:

**Figure supplement 1**. Dynamics of the continuous-time model.

First, when beliefs are uncertain (i.e., regimes around $L_{n-1} = 0$ in *Figures 1B, 2A,C*; *Equation 3a*, in which $K_n$ predominates over $\theta_n$), the model acts like a leaky accumulator, in which the prior expectation is a fraction of the previous belief (*Busemeyer and Townsend, 1993*; *Usher and McClelland, 2001*; *Bogacz et al., 2006*; *Tsetsos et al., 2012*). Thus, the dynamics of a leaky accumulator can, in principle, act like the normative model, but only in the low-certainty regime (*Figure 3*). In this regime, the normative leak is adaptive, which has been demonstrated previously (*Ossmy et al., 2013*), and is directly dependent on *H*, which has not been described previously. For low *H* and thus relative stability, a small leak provides long integration times. For $H \approx 0.5$ (the correct answer is equally likely to stay or

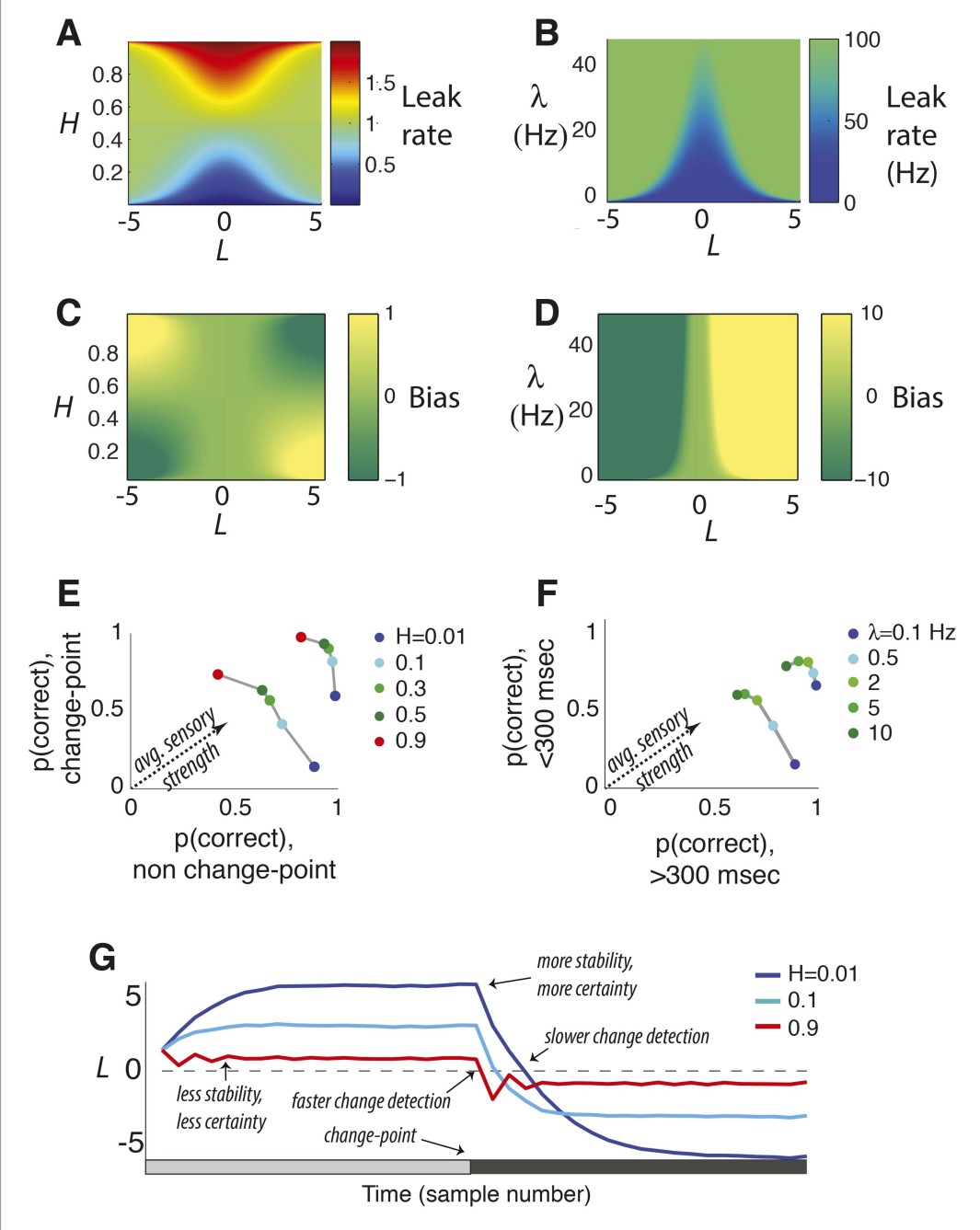

**Figure 2**. Features of the discrete-time (**A**, **C**, **E**, **G**) and continuous-time (**B**, **D**, **F**) normative models. (**A**, **B**) Leak rate as a function belief state and hazard rate. Blues are the least leaky and correspond to longer temporal accumulation; reds are the most leaky and correspond to a sign reversal in the change in current belief, resulting in damped oscillations in choice behavior. For the continuous-time model (**B**), there are no leak rates analogous to discrete-time $H > 0.5$. (**C**, **D**) Bias as a function belief state and hazard rate. Dark greens are the most biased in favor of the alternative associated with negative log-odds; yellows are the most biased in favor of the other alternative. (**E**, **F**) Predicted choice accuracy one sample (**E**) or <300 ms (**F**) after a change-point vs during steady-state conditions, at different expected hazard rates, as shown, for two difference strengths of evidence (**E**: $|LLR| = 0.5$ for leftmost curves and 5 for rightmost; **F**: $|LLR| = 4/s$ for leftmost curves and 80/s for rightmost). (**G**) Average belief from the discrete-time model over 1000 simulations for each condition shown, each with a single change-point at trial 20.

switch after each sample), the model discards all historical information and $L$ depends only on $LLR$. For $H > 0.5$ (the correct answer is more likely to switch after each sample), the prior expectation undergoes damped oscillations (*Figure 2G*), even when the source of evidence is transiently stable. These oscillations repeatedly switch the direction of existing beliefs because of the high expected probability of change on each discrete time step.

Second, as the magnitude of $L_{n-1}$ increases and belief certainty becomes high (i.e., regimes around $L_{n-1}$ far from zero in *Figures 1B, 2A,C*; *Equation 3b,c*, in which $\theta_n$ predominates over $K_n$), such as when the incoming evidence is strong or during periods of stability in the source, the prior expectation approaches a 'stabilizing boundary' whose height directly depends on $H$. Thus, the dynamics of a model that stabilizes the decision process at a hazard-dependent value can, in principle, act like the normative model, but only in the high-certainty regime (*Figure 3*). This boundary represents a suspension of the accumulation process but, unlike the decision bound in the SPRT and related models (*Barnard, 1946*; *Wald, 1947*; *Good, 1979*; *Link, 1992*; *Smith and Ratcliff, 2004*; *Bogacz et al., 2006*; *Gold and Shadlen, 2007*), does not terminate the decision process. Instead, it stabilizes $L_n$ when no changes occur (i.e., temporarily ending further evidence accumulation) while still allowing for the sampling of new evidence that might lead to changes in belief and a re-start of the accumulation process (*Resulaj et al., 2009*). The stabilizing boundary is also in contrast to the asymptote in leaky accumulation, which increases linearly with the strength of evidence (*Busemeyer and Townsend, 1993*; *Usher and McClelland, 2001*; *Bogacz et al., 2006*; *Tsetsos et al., 2012*).

Together these properties navigate an inherent trade-off between identification of steady signals and change detection. This trade-off depends on both evidence strength and expected $H$ (*Figure 2E,G*). For weak evidence, the trade-off is most severe, as the model uses expected $H$ to err on the side of either detecting changes quickly when $H$ is high or identifying stable signals when $H$ is low. As the strength of evidence increases, performance improves steadily for both conditions and the trade-off diminishes.

## Continuous-time version

We also developed a continuous-time version of the model (*Figure 1C*) that allowed for a more direct comparison to drift-diffusion and other continuous-time models of decision-making (*Figure 1C*) (*Smith and Ratcliff, 2004*; *Bogacz et al., 2006*; *Gold and Shadlen, 2007*). The model is based on the optimal filter for a Markov jump process with two states and stationary white-noise emissions (*Zakai, 1965*; *Liptser and Shiryaev, 1977*; *Crisan and Rozovskii, 2011*). Here we write the incoming evidence as a continuous-time sequence of noisy observations: $dx(t) = h(t)d(t) + \sigma dW$, where $h(t) = \pm\mu$, with the sign depending on which source is generating data at time $t$, and $\sigma$ is the standard deviation of the noise in a standard Wiener process $dW$. The source $h(t)$ jumps between states at an average rate $\lambda$, with jumps occurring as a Poisson process. Letting $A = 2\mu/\sigma^2$:

$$dL(t) = -2\lambda \sinh L(t)dt + Adx(t). \tag{4}$$

The result can be viewed as a nonlinear filter for the incoming evidence that is more general than the perfect or leaky integration central to previous models of decision-making between two alternatives (*Busemeyer and Townsend, 1993*; *Usher and McClelland, 2001*; *Bogacz et al., 2006*). In the special case that $\lambda = 0$, $dL(t) = Adx(t)$, which is perfect integration of the noisy observations $dx(t)$. Approximations of this model are similar to those for the discrete-time model (*Figure 2B,D*). When beliefs are uncertain ($L \approx 0$), $dL \approx -2\lambda Ldt + Adx(t)$, which results in an Ornstein-Uhlenbeck process over periods in which the source is perfectly stable (*Busemeyer and Townsend, 1993*; *Bogacz et al., 2006*). As certainty increases ($|L| > 0$), a simultaneous increase in leak rate and bias drives the decision variable to a stabilizing boundary (*Figure 1C*) with a probability distribution that has a heavy tail, reflecting dynamics that facilitate the detection of subsequent changes (*Figure 1—figure supplement 1*). As with the discrete-time model, these dynamics navigate the trade-off between identification of steady signals and change detection in a way that depends on both evidence strength and expected $\lambda$ (*Figure 2F*).

## Psychophysics

We used two separate tasks to investigate if and how human subjects could use these dynamics to adapt to different rates of change and find the optimal trade-off between stable signal identification and change detection. For both tasks, we found that: (1) subjects adapted, albeit imperfectly, to

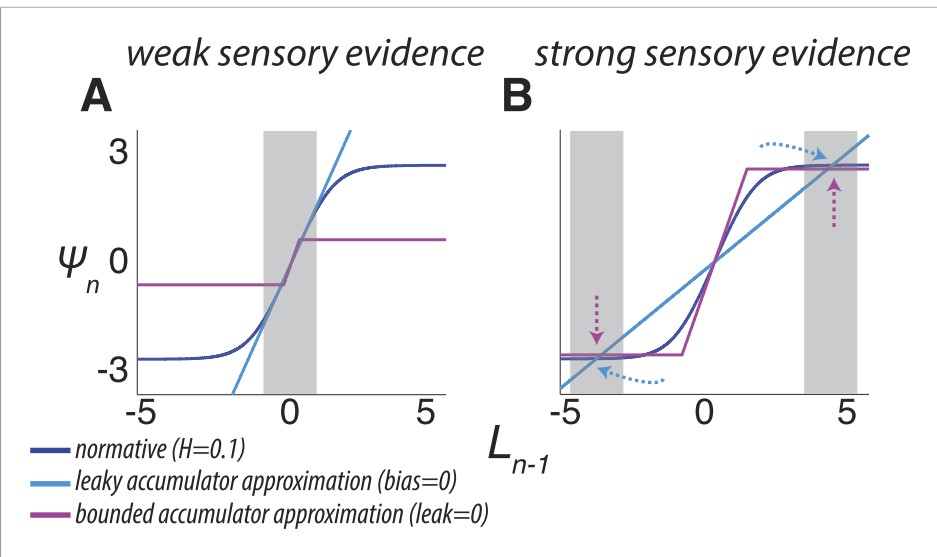

**Figure 3**. Two ways of approximating the discrete-time normative model, accounting separately for its dynamics when the sensory evidence is consistently weak (**A**, average I$LLR$I ≈ 0.25) or strong (**B**, average I$LLR$I ≈ 10). As in **Figure 1B**, each panel has discrete-time log-prior odds as a function of the belief at the previous moment. Dark blue lines correspond to the normative model for $H = 0.1$. Light blue lines correspond to a leaky accumulator with no bias, related to the linear approximation in **Equation 3a** but optimized to best approximate the normative model separately for each average evidence strength in **A** and **B**. Magenta lines correspond to perfect accumulation (no leak) to a stabilizing boundary related to **Equation 3b,c**, also optimized for each evidence strength. In general, the leaky accumulator is better at approximating the normative solution for weak sensory evidence (**A**), whereas the bounded accumulator is better at approximating the solution for strong sensory evidence (**B**).

different hazard rates (via comparisons to a suboptimal model, which ignored block-wise changes in $H$) and used their subjective estimates of hazard rate in a manner consistent with the normative model; and (2) their choice dynamics were better described by the normative model than two other adaptive, but suboptimal, alternatives inspired by the approximations to the normative model (one was an accumulator with a leak that could vary as a free parameter for each hazard-specific block of trials but no stabilizing boundary; the other was a perfect accumulator with a stabilizing boundary that could vary as a free parameter for each hazard-specific block of trials; see **Figure 3**).

## 'Triangles' task

This task required subjects to make trial-by-trial choices about which of two spatially separated triangles on a computer screen was the source of a single data point presented on that trial, represented as the position of a star on the screen (**Figure 4A,B**). Subjects could thus make choices based on accumulated evidence after each new sample of data, with the $LLR$ for each star corresponding to its position relative to each triangle. The correct source changed at a hazard rate that was constant within a block of trials but varied across blocks (0.05–0.95). In a subset of sessions (65 of 111), learning was facilitated by beginning and ending each block of trials with stretches of trial-by-trial feedback about the correct answer. However, subjects were never instructed on what the hazard rates were or when they would change.

The subjects were able to adapt their decision-making to these different hazard rates, as assessed by direct fits of their choice data by the normative model. Specifically, models that allowed subjective $H$ to freely vary by block provided better fits to the data than a suboptimal model that ignored the block-wise changes in objective $H$ (median ± bootstrapped SEM difference of Bayesian Information Criterion, or BIC, from normative vs block-independent $H$ fits was −23.721 ± 9.671, Wilcoxon signed-rank test, p < 0.0001; per subject, the normative fits were better in 43 of 48 subjects using a signed-rank test, Bonferroni corrected p < 0.05). Overall, the normative model performed well, with choice residuals centered around zero (mean ± std deviance residual = 0.003 ± 0.458) and

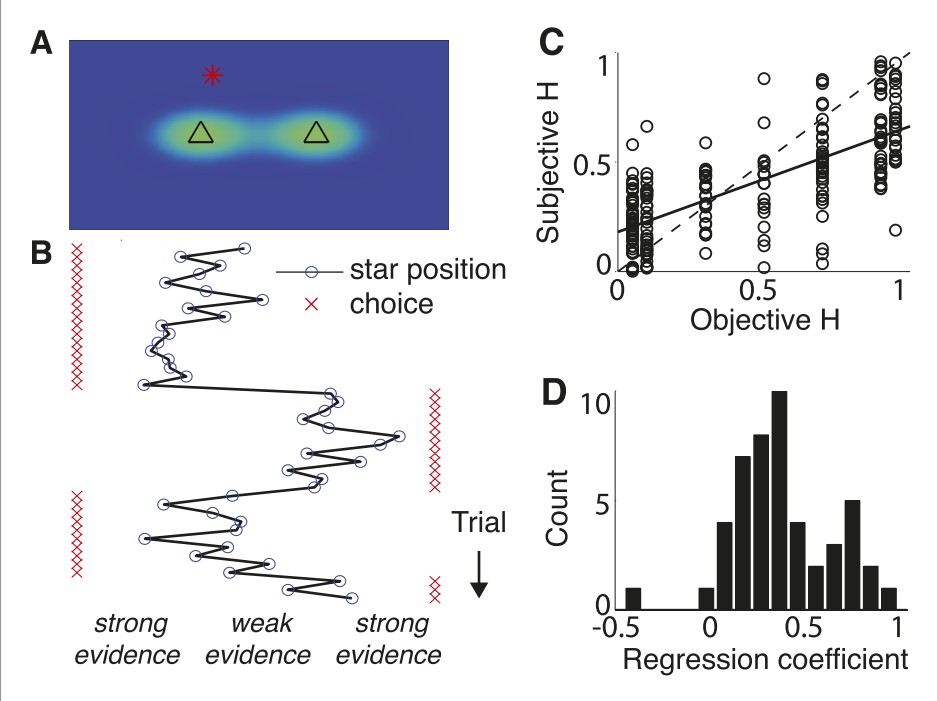

**Figure 4**. Triangles task and normative model fits. (**A**) Example task screen. Triangles and surrounding greenish clouds represent the means and variances of the two generative processes; red star is a single sample (in this case generated by the left process). (**B**) Sample trials, with actual star position indicated by blue circles and subject choices indicated by red 'x'. Star positions close to the center represent weak evidence for either of the alternatives because the respective probabilities of either source generating the star position are close. Star positions towards the edge of the screen represent strong evidence for the triangle to which the star is closest. (**C**) Block-wise subjective (fit) *H* vs objective *H*. Dotted line is unity; solid line is a least-squares fit. (**D**) Histogram of slope coefficients from least-squares fits as in **C**, calculated for individual subjects.

The following figure supplement is available for figure 4:

**Figure supplement 1**. Two examples of subjects adapting to objective hazard rate across an entire experimental session.

a reasonably close match to the choice data (median ± bootstrapped SEM McFadden's $r^2$ of 0.895 ± 0.016 across subjects). Moreover, the estimated values of subjective *H* from these fits were strongly correlated with objective *H* across all subjects (Pearson's $r = 0.721$, $p < 0.0001$; *Figure 4C*), with 47 of 48 individual subjects showing a regression slope of subjective on objective *H* that was >0 and in 46 of those cases was also <1 (*Figure 4D*, median ± bootstrapped SEM regression slope = 0.402 ± 0.042). However, although the subjects adapted their decision-making behavior appropriately for different values of *H*, their subjective estimates of *H* tended towards *H* ~ 0.5, for which the history of evidence is irrelevant. This tendency to mis-estimate extreme hazard rates did not appear to reflect insufficient learning opportunities, because these trends persisted even when restricting fits to the last 200 trials of each block (across subjects and blocks, bootstrapped regression slope of subjective on objective *H* = 0.373 ± 0.044; see also *Figure 4—figure supplement 1*) or to blocks beginning with explicit, trial-to-trial feedback (regression slope = 0.562 ± 0.038).

The subjects appeared to be using these learned, subjective estimates of hazard rate in a manner consistent with the normative model, for several reasons. First, their choice dynamics directly reflected biases predicted by the two regimes of the normative model (*Equation 3*, *Figure 5A–F*). When certainty was high (i.e., choices just following a trial in which the star position was far to the left or right), the subjects consistently showed hazard-dependent biases that were predicted by the stabilizing boundary of the model: weak evidence interpreted as stability for low *H* and change for high *H* (Spearman's $r = 0.890$, $p < 0.0001$, comparing predicted and actual biases from individual

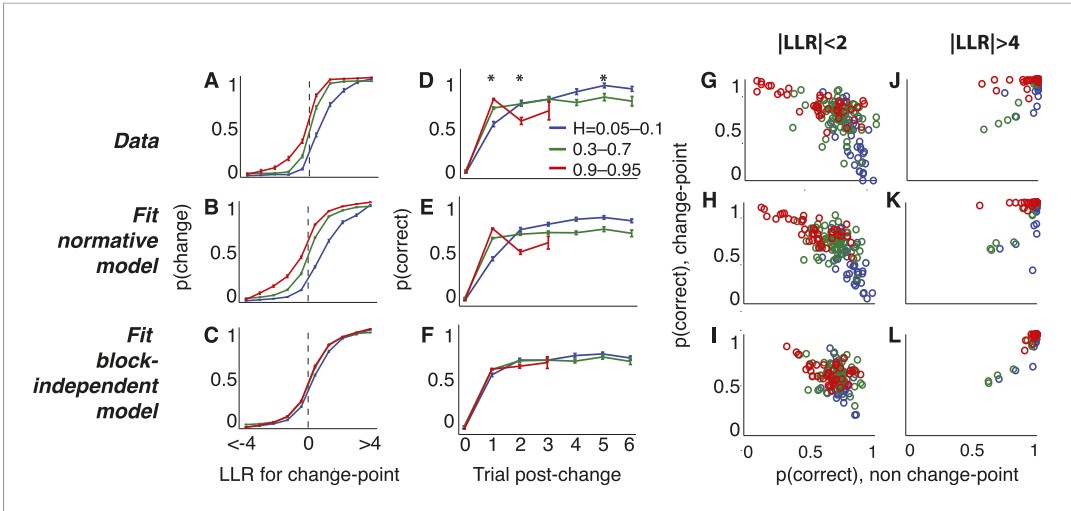

**Figure 5**. Triangles task choice data pooled across all 48 subjects (top row), plus predictions from fits to the normative model allowing a different subjective hazard rate to be assigned to each block (middle row) and from fits to a model with subjective hazard rates randomly assigned across trials (bottom row). Colors are different ranges of objective *H*, as shown in panel **D**. Errorbars are bootstrapped sem. (**A–C**) Probability of switching choices as a function of the *LLR* for a change in the correct answer. Data were restricted to trials following strong evidence (|LLR| > 4) to directly investigate the 'strong belief' regime of the model predictions. (**D–F**) Probability of switching sides in which a strong *LLR* (|LLR| > 4) for the original side was followed by a change-point and weak (|LLR| < 2) evidence for the opposite side. Significant differences by *H* in subject data are indicated by asterisks (Bonferroni corrected p < 0.05, $\chi^2$ test). (**G–L**) Block-by-block choice accuracy on change-point vs non-change-point trials when the evidence (magnitude of *LLR*, as indicated) was relatively weak (**G–I**) or strong (**J–L**). Points are individual blocks that included ≥5 trials with the indicated conditions.

blocks; *Figure 5A,B*). When weak evidence persisted without change-points for a run of trials following a change-point, choice dynamics consistently reflected the hazard-dependent leak predicted by the model (*Figure 2G*, *Figure 5D,E*): gradual updates for low subjective *H*, immediate updates for subjective *H* ≈ 0.5, and damped oscillations for high subjective *H* (including lower accuracy on two vs one trial following the change-point). Fits to the block-independent model did not show these *H*-dependent choice dynamics, confirming that the choice dynamics did not result from any differences in how the randomly generated stars were sampled under the different conditions used for these analyses (*Figure 5C,F*).

Second, the subjects' decision processes reflected a strong, hazard- and evidence-strength-dependent trade-off between detecting changes and identifying steady signals, as predicted by the normative model (*Figure 5G–L*). When the evidence was weak (star positions close to the midline), accuracy following a change-point was highest for high *H* (mean ± bootstrapped SEM across blocks = 73.7 ± 2.4% correct) then declined steadily for intermediate (66.4 ± 1.2%) and low *H* (50.0 ± 7.3%; Spearman's correlation between change-point accuracy and *H* was 0.559, p < 0.0001, vs a predicted correlation from the normative model of 0.651). In contrast, accuracy on non change-point trials with the same strength of evidence was lowest for high *H* (57.1 ± 4.1%) then improved steadily for intermediate (71.4 ± 1.2%) and low *H* (81.1 ± 2.2%; Spearman's correlation between non change-point accuracy and *H* was −0.502, p < 0.0001, vs a normative prediction of −0.625; *Figure 5G,H*). When the evidence was strong, the trade-off was much smaller, as predicted by the normative model (Spearman's correlation between change-point accuracy and *H* was −0.040, p = 0.557 and between non change-point accuracy and *H* was −0.005, p = 0.945; normative predictions were 0.027 and 0.010, respectively; *Figure 5J,K*). Fits to the block-independent model did not show these *H*-dependent trade-offs, for either weak or strong evidence (*Figure 5I,L*).

Third, we used choice data to directly estimate the mapping of subjective beliefs to priors ($L_{n-1}$ to $\Psi_n$, *Figure 6*; compare to *Figure 1B*). Like for the normative model, the estimated mappings depended on subjective *H* (one-way MANOVA for the groups shown in *Figure 6A*, p < 0.0001). Moreover, for

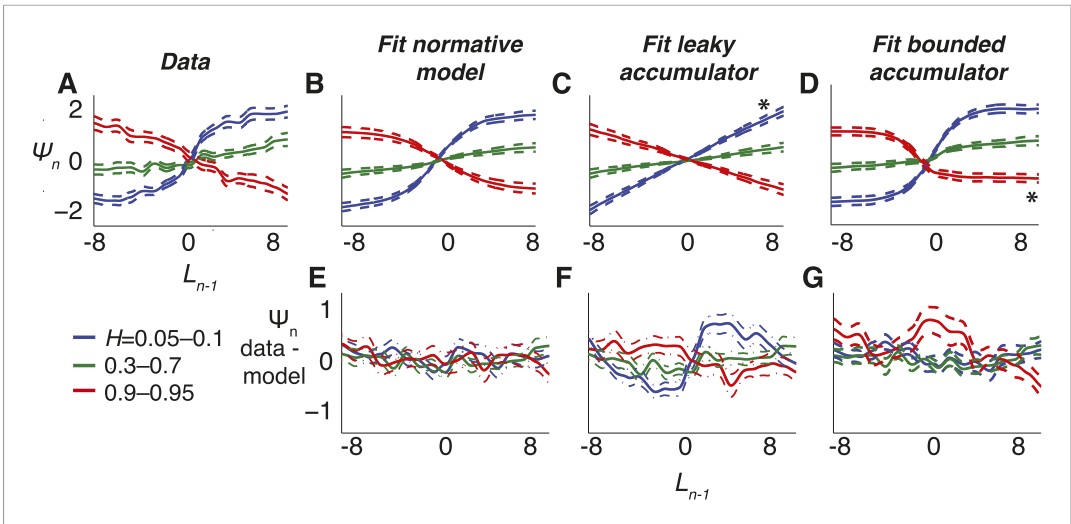

**Figure 6**. Belief dynamics estimated directly from the data and compared to predictions from the normative model and two suboptimal approximations (*Figure 3*). (**A–D**) Estimates of the log prior-odds on a given trial as a function of belief on the previous trial (compare to *Figure 1B*) computed for each experimental block, grouped by objective *H*, as indicated in the legend below panel **A**. Solid lines are across-block means, and dashed lines are sem. (**A**) Data. (**B**) Fit normative model. (**C**) Fit hazard-dependent leaky accumulator. (**D**) Fit model with perfect accumulation to a hazard-dependent stabilizing boundary. Asterisks in panels **C** and **D** indicate hazard-rate regimes in which estimates from the corresponding model prediction differed significantly from data estimates using Hotelling's *t*-test with a Bonferroni corrected p < 0.05. (**E–G**) Hazard-specific differences between the data estimates and model predictions.

each *H*-group, these mappings matched predictions of the normative model (*Figure 6B,E*; Hotelling's *t*-test comparing data and model, p = 0.189, 0.321, and 0.086 for low, medium, and high values of objective *H*, respectively).

In contrast, the choice data from the triangles task were not as well matched by either of the two adaptive, suboptimal models we considered (*Figure 3, 6*). The leaky-accumulator model had worse overall fits to the choice data than the normative model for 34 of 48 subjects (median ± SEM difference in BIC = −5.179 ± 1.967, Wilcoxon signed-rank test, p < 0.0001) and predicted mappings of subjective beliefs to priors that matched the pooled data only for medium and high values of *H* but not for low values of *H*, which lacked the asymptotic regime prescribed by the normative model when beliefs were more certain (*Figure 6C,F*; Hotelling's t-test, p < 0.0001 for low objective *H*, and p = 0.312 and 0.545 for medium and high objective *H*, respectively). Likewise, the model with perfect accumulation to a hazard-specific stabilizing boundary had worse overall fits to the choice data than the normative model for 34 of 48 subjects (−0.942 ± 0.487, p = 0.007), reflecting the lack of a leaky-accumulation regime prescribed by the normative model when beliefs were uncertain and *H* was high (*Figure 6D,G*; Hotelling's t-test, p = 0.228, 0.463, and 0.017, respectively). This relatively modest, but reliable, difference in BIC reflected an inherent difficulty in distinguishing these models with the particular task conditions we used (fitting simulated data from either model yielded similarly small BIC differences: −1.448 ± 0.900 for simulations based on the normative fits and 1.393 ± 1.054 for simulations using the stabilizing-boundary fits). Both suboptimal models had best-fitting, subjective hazard rates that, even more than for the normative fits, tended to overestimate small objective values and underestimate large objective values, further supporting the idea that the subjects were using mis-estimated hazard rates to make their decisions (regression slope of subjective vs objective *H* = 0.115 ± 0.023 for the leaky accumulator and 0.290 ± 0.044 for the perfect accumulator to a stabilizing boundary, p < 0.0001 when compared to slopes from the normative fits in both cases).

## Dots-reversal task

This task was a novel version of a commonly used random-dot motion task (*Britten et al., 1992*). For this 'dots-reversal' task, the direction of coherent motion underwent sudden changes within trials.

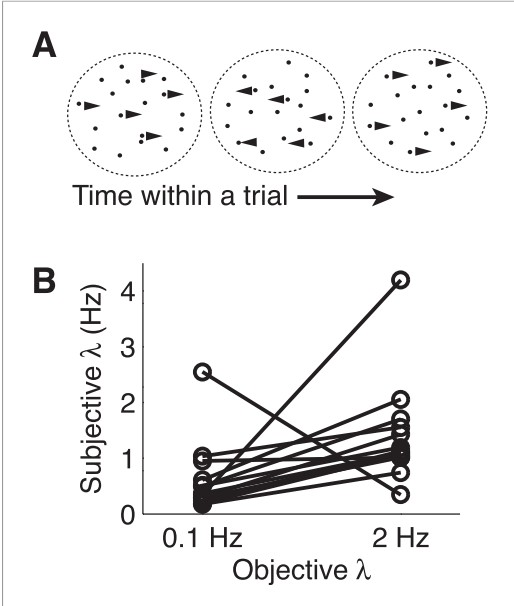

**Figure 7**. Dots-reversal task and normative model fits. (**A**) Representation of a reversing-dots stimulus for a single trial. The subject was instructed to indicate the final, perceived direction of motion. (**B**) Subjective hazard rate, estimated from direct fits of choice data by the normative model with hazard rate as a free parameter, plotted as a function of objective hazard rate. Each pair of connected points represents data from an individual subject.

Each subject participated in two separate sessions, one in which changes occurred at a relatively slow rate (0.1 Hz), and one in which changes occurred at a fast rate (2.0 Hz) rate (*Figure 7*, *Videos 1–4*). Motion strength (coherence) was fixed to either a high or low value within each trial, and subjects were instructed to pay attention to the stimulus throughout the trial and then indicate its final direction, after which they received feedback on the correct answer.

As with the triangles task, the subjects were able to adapt their decision-making to these different hazard rates. Models that allowed subjective $\lambda$ (change-point rate, here treated as a continuous-time variable) to vary with objective $\lambda$ provided better fits to the data than a model that ignored the session-specific changes in $\lambda$ (median $\pm$ SEM difference in BIC from the normative vs block-independent model fits was −4.790 $\pm$ 1.216, p < 0.0005; per subject, normative BIC values were significantly lower in 9 of 13 subjects with a Bonferroni corrected p < 0.05, Wilcoxon signed-rank test). The normative model performed well overall, with choice residuals centered around zero (mean $\pm$ std deviance residual = 0.053 $\pm$ 0.897) and a reasonably close match to the choice data (median $\pm$ bootstrapped SEM McFadden $r^2$ of 0.385 $\pm$ 0.055 across subjects). Of the 13 subjects, 12 had best-fitting values of adaptive, subjective $\lambda$ that showed appropriate sensitivity to objective $\lambda$, with estimated subjective $\lambda$ lower on 0.1 vs 2.0 Hz trials (*Figure 7B*; Wilcoxon signed-rank p < 0.05). However, like for the triangles task, the subjects tended to overestimate low values and underestimate high values of $\lambda$ (median $\pm$ bootstrapped SEM estimated subjective $\lambda$ = 0.365 $\pm$ 0.109 and 1.129 $\pm$ 0.168 Hz for the 0.1-Hz and 2-Hz conditions respectively), with a similar tendency even when restricting model fits to the last 50 trials of each session to account for learning (subjective $\lambda$ = 0.291 $\pm$ 0.122 and 0.968 $\pm$ 0.207 Hz for the 0.1-Hz and 2-Hz conditions, respectively).

Also consistent with our results from the triangles task, the subjects appeared to be using their adaptive estimates of $\lambda$ in a manner consistent with the normative model, based on several lines of evidence. First, the choice data exhibited dynamics predicted by the normative model (*Figure 8*). For high-coherence trials, the strong sensory evidence dominated the decision process, yielding >90% accuracy within 500 ms following the final change in direction irrespective of the rate of preceding direction changes (*Figure 8D,E*). In contrast, for low-coherence trials, integration times were strongly dependent on hazard rate (i.e., greater effects of $\psi$ in *Equation 1*) Specifically, accuracy improved more steeply as a function of viewing duration for the low- vs high-hazard condition: performance was worse for the 0.1-Hz condition for durations <500 ms, reflecting persistence of the perceived direction of motion just prior to the final change-point (i.e., direction reversal), but rose as viewing duration increased and exceeded performance for the 2-Hz condition at long durations (*Figure 8A,B*). These dynamics, particularly at low coherences, were not predicted by the block-independent model and thus did not reflect uneven sampling of the data under the different coherence and hazard conditions (*Figure 8C,F*).

Second, as with the triangles task, this decision process reflected a strong hazard- and evidence-strength-dependent trade-off between detecting changes and identifying steady signals, as predicted by the normative model (*Figure 8G–K*). For low-coherence trials, choice accuracy was lower for the low-hazard condition just following a change-point (median $\pm$ bootstrapped SEM 7.1 $\pm$ 11.1% and

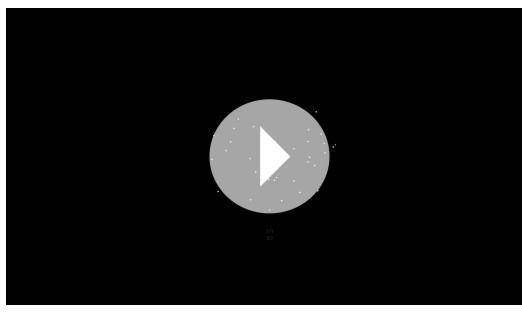

**Video 1.** Example random dot motion stimulus with
0.1 Hz changes, at 80% ('high') coherence.

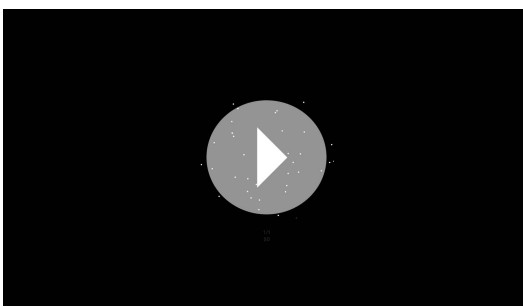

**Video 2.** Example random dot motion stimulus with
0.1 Hz changes, at 20% ('low') coherence.

40.3 ± 2.8%, for low- and high-hazard sessions respectively, when the stimulus was shown for <300 ms following the final change-point, Wilcoxon signed rank, p < 0.01, vs predicted accuracies of 34.6 ± 4.7% and 44.9 ± 3.1%, respectively) but was greater for the low-hazard condition thereafter (83.6 ± 3.6% and 69.9 ± 8.3%, respectively, p < 0.0005, vs predicted 86.2 ± 2.3% and 74.2 ± 2.9%, respectively). For high-coherence trials, choice accuracy was much higher overall and, consistent with the normative model, showed a weaker trade-off, with no difference by hazard-rate condition for short viewing durations following the final change point (66.7 ± 14.6% and 66.1 ± 7.3% for 0.1-Hz and 2-Hz sessions respectively, p = 0.831, vs predicted 46.9 ± 9.4% and 59.7 ± 3.7%, respectively) and only a slight difference for longer post-change durations (100 ± 0% and 93.5 ± 1.4%, respectively, p = 0.004, vs predicted 97.8 ± 0.4% and 92.1 ± 2.0%, respectively). In contrast, the block-independent model did not predict these hazard-dependent trade-offs (*Figure 8I,L*).

Third, we directly measured the dependence of choice dynamics on both the hazard rate and the strength of the sensory evidence by fitting choice data to integrating models with separate leaks for each hazard-specific session and coherence level (*Figure 9*). As predicted by the normative model (*Figures 3A, 8B,E*), the best-fitting leak depended on both (Friedman test, p < 0.0005 for the effect of hazard rate and p < 0.0001 for the effect of motion coherence; *Figure 9A*). The per-subject, pairwise differences between best-fitting leak, computed with respect to either coherence or hazard rate, did not differ from predictions of the normative model (median ± bootstrapped SEM normalized data–model difference by hazard rate = −0.003 ± 0.021, by coherence = −0.001 ± 0.050, Wilcoxon signed rank p = 0.501 and 0.292, respectively).

In contrast, the choice data from the dots-reversal task were not as well matched by either of the two adaptive, suboptimal models we considered (*Figure 3*). The leaky-integrator model, in which the leak depended on hazard rate but not coherence level, had worse overall fits to the choice data than the normative model for 10 of 13 subjects (median ± bootstrapped SEM difference in BIC = −4.066 ± 3.078, Wilcoxon signed rank, p < 0.05), failing in particular to capture the strongly coherence-dependent leak of the data and the normative model (*Figure 9C,F,H,I*; normalized data–model median ± bootstrapped SEM difference between change in leak by coherence = 0.241 ± 0.067, Wilcoxon signed rank p < 0.0005). The normative model also outperformed the model with perfect integration to a stabilizing boundary that freely varied by objective hazard rate (BIC was lower for 9 of 13 subjects, −4.305 ± 2.662, p < 0.05). This suboptimal model had a more subtle deviation from the data, consisting primarily of an exaggerated dependence of leak on coherence (*Figure 9D,G,J,K*; per-subject, normalized data–model difference between change in leak by coherence = −0.189 ± 0.059, Wilcoxon signed rank p < 0.0001). Like for the normative fits, both suboptimal models also had best-fitting subjective hazard rates that were imperfectly adapted to the objective values, further supporting the idea that the subjects were using imperfect estimates to make their decisions (best-fitting $\lambda$ = 0.789 ± 0.084 and 1.697 ± 0.227 Hz for the 0.1-Hz and 2-Hz conditions, respectively, for the leaky integrator and 4.378 ± 0.998 and 8.803 ± 1.159 Hz, respectively, for the perfect integrator to a stabilizing boundary).

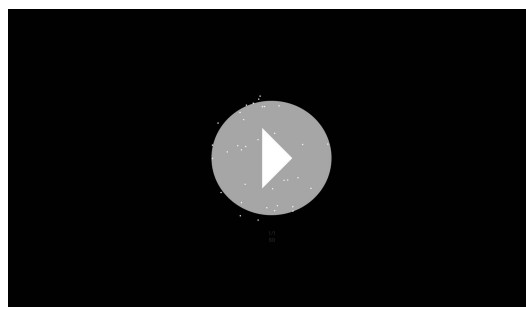

**Video 3.** Example random dot motion stimulus with 2 Hz changes, at 80% ('high') coherence.

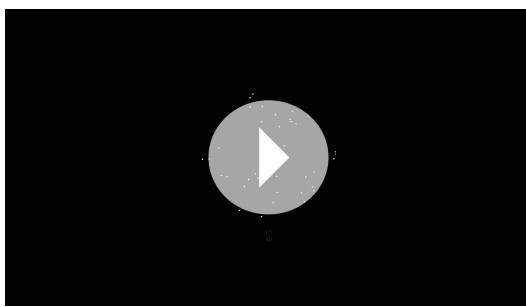

**Video 4.** Example random dot motion stimulus with 2 Hz changes, at 20% ('low') coherence.

## Discussion

We derived a normative model of evidence accumulation for decision tasks that is based on Bayesian principles for inferring changes in the statistics of a generative process (*Rabiner, 1989*; *Adams and MacKay, 2007*; *Behrens et al., 2007*; *Fearnhead and Liu, 2007*; *Brown and Steyvers, 2009*; *Wilson and Finkel, 2009*; *Nassar et al., 2010, 2012*; *Wilson et al., 2010*; *Boerlin et al., 2013*; *Wilson et al., 2013*; *Gonzalez Castro et al., 2014*; *McGuire et al., 2014*; *Sato and Kording, 2014*). Our model incorporates change detection into sequential-sampling decision models and is related to other, modified versions of these models that have been used to combine multiple sensory cues of different but known reliabilities or infer unknown sensory reliability assumed to be stable during the course of decision-making (*Hanks et al., 2011*; *Deneve, 2012*; *Drugowitsch et al., 2014*). However, unlike those models, which invoked a separate learning-rate term or had other, more complex forms, our model casts adaptation directly in the context of the evidence-accumulation process that is a key focus of studies of decision-making (*Usher and McClelland, 2001*; *Roitman and Shadlen, 2002*; *Huk and Shadlen, 2005*; *Uchida et al., 2006*; *Brunton et al., 2013*; *Hanks et al., 2015*). This formulation allowed us to identify, for the first time, features of evidence accumulation that can underlie normative, adaptive decision-making, including expectation-dependent changes in leaky accumulation when beliefs are weak and saturating accumulation when beliefs are stronger. We showed that human subjects made decisions on two separate tasks, requiring evidence accumulation either across or within trials, that were consistent with the adaptive, hazard-dependent accumulation process prescribed by the model.

Our findings substantially extend previous studies that similarly suggested that human decision-making behavior can reflect adaptations to the rate of environmental changes (*Behrens et al., 2007*; *Brown and Steyvers, 2009*; *Gonzalez Castro et al., 2014*). Specifically, we showed that subjects could both learn a range of hazard rates and then use those learned rates in a normative manner to interpret sequences of evidence to make decisions. However, they tended to learn imperfectly, over-estimating low hazard rates and under-estimating high hazard rates. Thus, although their use of these imperfectly learned hazard rates was consistent with the normative model, their overall decisions in some cases fell short of the ideal observer. Our framework provides a new way to interpret these deviations from optimality: not simply as poor performance, but rather as different, hazard-dependent set-points of an inherent trade-off. This tradeoff balances sensitivity to change during periods of expected instability, and sensitivity to steady-state signals during periods of expected stability. These different set points may have reflected certain prior expectations about the improbability of either perfect stability or excessive instability that could constrain performance when those conditions occur.

Such prior expectations about a lack of perfect environmental stability interpreted in the context of our framework might also provide new insights into previous studies of the temporal dynamics of evidence accumulation. In some cases, decisions about perfectly stable stimuli appear to involve perfect accumulation, as described by drift-diffusion and related models (*Gold and Shadlen, 2000*; *Roitman and Shadlen, 2002*; *Brunton et al., 2013*; *Hanks et al., 2015*). Under those conditions, deviations from perfect accumulation in the brain may be considered as inefficient, operating under other constraints

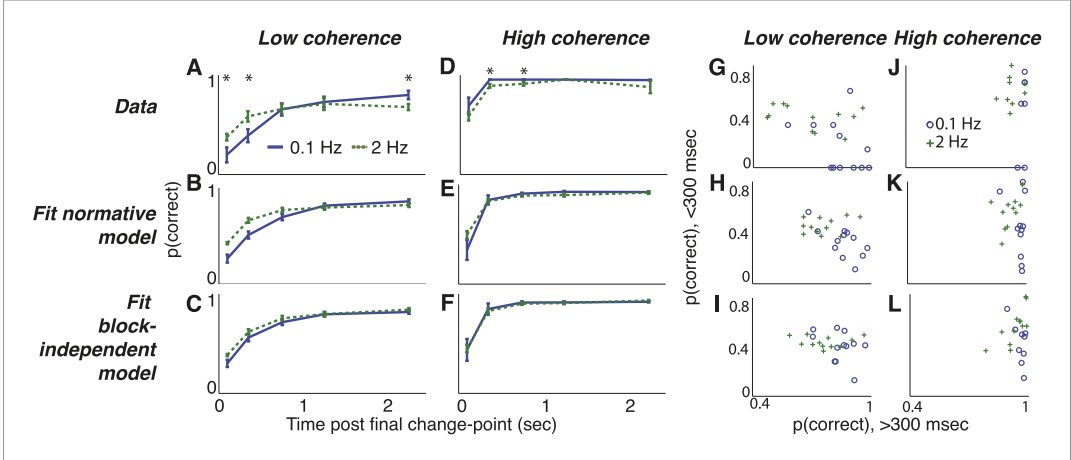

**Figure 8**. Dots-reversal task choice dynamics comparing the pooled data from 13 subjects (top row) with predictions from fits to the normative model (middle row) and to the normative model with the subjective hazard rates shuffled across trials and blocks for each subject and session (bottom row). (**A–F**) accuracy (±bootstrapped sem) as a function of viewing time following the final direction change within a trial for low- (**A–C**) and high- (**D–F**) coherence stimuli for the two hazard-rate conditions (indicated in panel **A**). Asterisks in panels **A** and **D** indicate a significant difference between the two hazard-rate conditions (bootstrapped t-test, Bonferroni corrected p < 0.05). (**G–L**) Dots-reversal task trade-off between accuracy on trials in which the final viewing duration was <300 ms vs >300 ms for different hazard rates (indicated in panel **J**), when the motion coherence was low (**G–I**) or high (**J–L**).

(e.g., computational costs), or of uncertain relevance to decision-making (*Usher and McClelland, 2001*; *Drugowitsch et al., 2012*). In contrast, our results imply that at least some deviations from perfect accumulation might reflect normative adjustments to expected instabilities, even under the nominally stable conditions used for many tasks. For example, leaky accumulation that places more emphasis on recent vs past information or rates of accumulation that vary as a function of time, which can account for the temporal dynamics of certain decisions about stimuli that are presented with stable statistics for 100's of ms or more, might reflect prior expectations that instabilities are likely to occur within that time frame (*Usher and McClelland, 2001*; *Eckhoff et al., 2008*). Likewise, reports of an 'urgency' signal that limits temporal integration based on a drive to respond quickly might reflect similar expectations of impending instabilities (*Reddi and Carpenter, 2000*; *Ditterich, 2006*; *Cisek et al., 2009*; *Drugowitsch et al., 2012*; *Thura et al., 2012*). More extreme expectations of instabilities might relate to other tasks that appear not to require temporal integration at all and instead show little dependence of performance on stimulus duration beyond what is needed to activate the sensory detectors (*Ludwig et al., 2005*; *Uchida et al., 2006*). These interpretations are consistent with the idea that the temporal integration window for many kinds of decisions might be highly flexible and adapt to the temporal dynamics of the environment (*Ossmy et al., 2013*; *Gonzalez Castro et al., 2014*). Insofar as the accumulated evidence that serves as the decision variable governing choice behavior can also be thought of as a confidence signal, such adaptive dynamics might also pertain to confidence judgments associated with certain decision tasks (*Kepecs et al., 2008*; *Kiani and Shadlen, 2009*; *Ma and Jazayeri, 2014*). Further work is needed to understand if and how these findings can be understood in the context of a common set of normative principles that balance the identification of steady signals with change detection.

Our results might also have implications for understanding the trade-off between speed and accuracy inherent to many tasks (*Gold and Shadlen, 2007*; *Bogacz et al., 2010*). Sequential-sampling models like drift-diffusion typically account for this trade-off in terms of an absorbing decision boundary. This boundary can be set to a pre-defined value to terminate the decision process while emphasizing either speed or accuracy at the expense of the other, or possibly balancing the two in the service of maximizing related quantities like reward rate (*Gold and Shadlen, 2002*; *Palmer et al., 2005*; *Bogacz et al., 2006*, *2010*; *Simen et al., 2009*). Alternatively, in our model the adaptive accumulation process can be suspended, at least temporarily, not by an extrinsically imposed decision rule like an absorbing decision boundary but rather by the non-linear dynamics of the accumulation process itself. In principle, certain

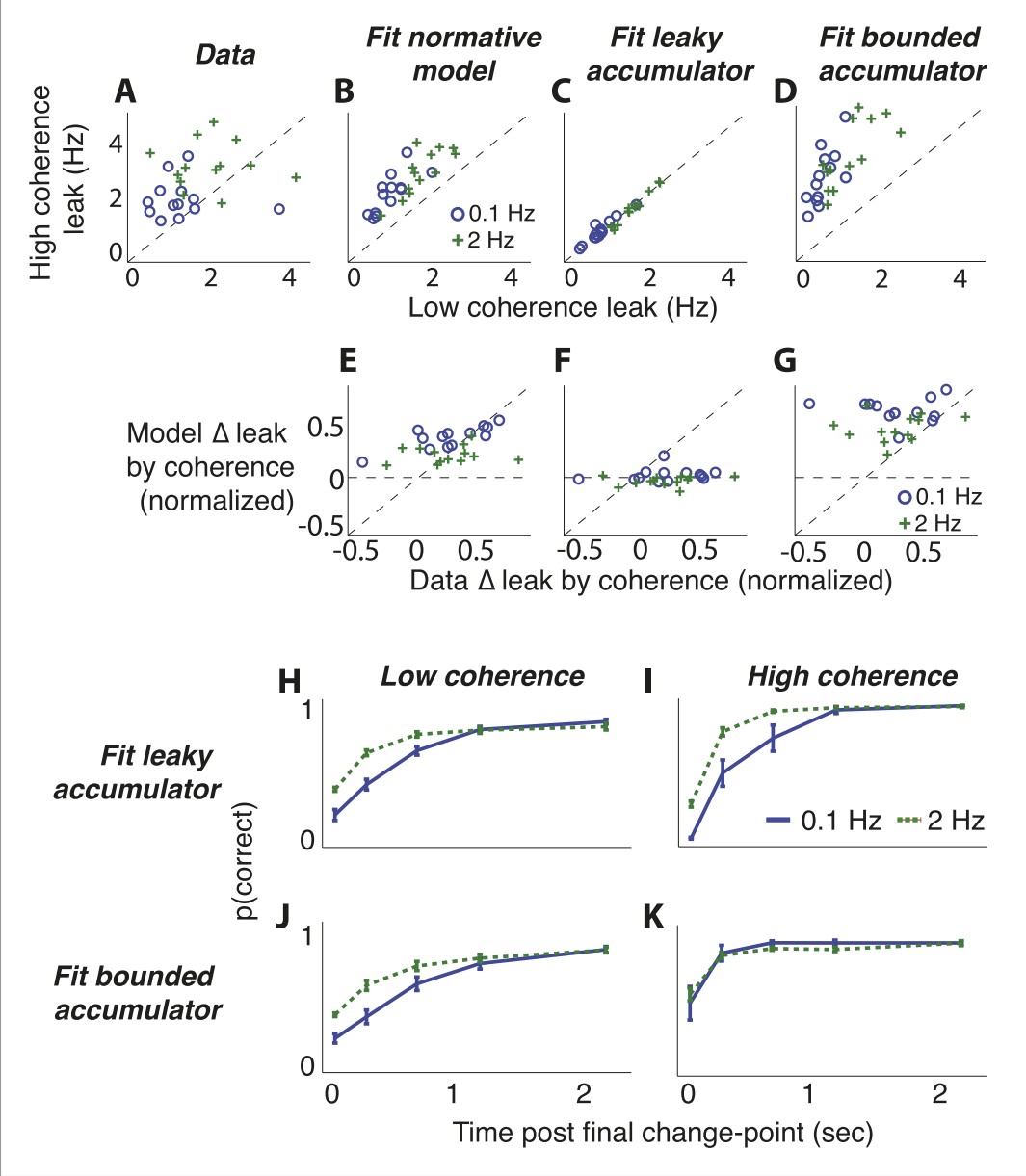

**Figure 9**. Comparison of predictions from normative model vs from the suboptimal approximations (*Figure 3*) for the dots-reversal task. (**A**–**D**) Parameter fits from a leaky-accumulator model with separate leak–rate parameters per hazard rate (as indicated in panel **B**) and coherence (yielding four leaks per subject). (**A**) Leaks fit to choice data. (**B**) Leaks fit to predicted choices from the normative model using best-fitting parameters from subject data. (**C**) Leaks fit to predicted choices from the leaky-accumulator model using leaks that depended on the session-specific hazard rate but not coherence. (**D**) Leaks fit to predicted choices from the bounded-accumulator model using boundaries that depended on the session-specific hazard rate but not coherence. (**E**–**G**) Difference in the best-fitting leak to the two coherences predicted by each of the models above plotted against the difference in leaks from the direct fits to the choice data, separated by hazard rate (see panel **B**). Differences are normalized by sum of leaks from each of the coherences. In **A**–**G**, each point represents a single subject and hazard-rate condition. (**H**–**K**) Predicted accuracy (±bootstrapped sem) as a function of viewing time following the final direction change within a trial for low- (**H**, **J**) and high- (**I**, **K**) coherence stimuli for the two hazard-rate conditions (indicated in panel **I**), calculated as in *Figure 8* but for predictions by fits to the leaky- (**H**, **I**) and bounded- (**J**, **K**) accumulator models.

decisions might be made by committing to an alternative once this asymptotic regime is reached. This regime represents an upper limit on the expected level of confidence and thus precludes the need for either additional data for that alternative or for an additional boundary. In this case, the resulting speed-accuracy trade-off would not necessarily reflect a pre-defined attempt to control those factors explicitly but rather expectations about the rate at which the evidence-generating process is changing.

Future work is needed to investigate how key features of our model might be implemented in the nervous system for different tasks and different timescales. Previous studies using tasks that required information accumulation on the timescale of the triangles task (e.g., over many seconds to minutes) have similarly suggested that humans can approximate optimal change detection, which in some cases includes a sensitivity to different hazard rates (*Behrens et al., 2007*; *Brown and Steyvers, 2009*; *Nassar et al., 2010*). The neural mechanisms of these abilities are not yet known, but fMRI and pupillometry data suggest possible roles for the arousal system including the anterior cingulate cortex and the noradrenergic system, and genotype data imply possible contributions of the dopamine system (*Yu and Dayan, 2005*; *Nassar et al., 2012*; *Behrens et al., 2007*; *Krugel et al., 2009*; *O'Reilly et al., 2013*; *McGuire et al., 2014*). Conversely, evidence-accumulation processes that operate over shorter timescales, like for various versions of the random-dot motion task, have focused on dynamic neural signals in other parts of cortex, the basal ganglia, and the superior colliculus that can reflect the rapid build-up of evidence to select a particular motor response (in these cases involving eye movements) (*Gold and Shadlen, 2007*; *Ding and Gold, 2013*). There are some suggestions that these systems may interact under certain conditions (*O'Reilly et al., 2013*), but much more work is needed to understand the brain mechanisms responsible for the kinds of normative, scale-invariant dynamics of evidence accumulation we characterized in this study. Extending our framework to more than two alternatives and to conditions in which the statistics of the evidence changes gradually, as opposed to abruptly, would also be an important step towards better understanding how the brain accumulates and interprets dynamic evidence to solve complex, real-world problems.

## Materials and methods

### Discrete-time model

The normative model is based on the posterior probability each of option ($z_1$ or $z_2$) given all of the evidence collected so far ($x_{1:n}$), $q(z_{in}) \equiv p(z_{in}|x_{1:n})$. We assume that at each time step, there is a probability ($H$, for 'hazard rate') that there will be a switch in the correct option. Beginning with Bayes' Rule, and using the sum and product rules of probability, it can be shown that:

$$q(z_{1n}) \propto p(x_n|z_1)\left[(1-H)q(z_{1,n-1}) + Hq(z_{2,n-1})\right],$$
$$q(z_{2n}) \propto p(x_n|z_2)\left[Hq(z_{1,n-1}) + (1-H)q(z_{2,n-1})\right],$$

(5)

where $p(x_n|z_i)$ is the likelihood of observing the evidence from source $i$. This relationship is the forward recursion for the Baum-Welch algorithm in Hidden Markov Models and has been proven elsewhere (*Bishop, 2006*). We derived the model (*Equations 1, 2*) by taking the logarithm of the ratio of the two equations; that is, defining $L_n \equiv \log(q(z_{1n})/q(z_{2n}))$ and expanding the logarithm, giving:

$$L_n = \log[p(x_n|z_1)/p(x_n|z_2)] + \log\left[\left((1-H)q(z_{1,n-1}) + Hq(z_{2,n-1})\right)/\left(Hq(z_{1,n-1}) + (1-H)q(z_{2,n-1})\right)\right],$$

where the first term of the RHS is the *LLR* in *Equation 1* by definition. The second term of the RHS can be manipulated to yield $\psi$ (*Equation 2*) first by dividing both the numerator and denominator by $Hq(z_{2,n-1})$, then expanding the expression while using $\frac{q(z_{1,n-1})}{q(z_{2,n-1})} = \exp(L_{n-1})$ by definition, giving $\psi(L_{n-1}, H) = \log\left[\frac{1-H}{H}\exp(L_{n-1}) + 1\right] - \log\left[\exp(L_{n-1}) + \frac{1-H}{H}\right]$. Factoring out $\exp(L_{n-1})$ from the first term of the RHS yields *Equation 2*.

The special cases of $H = 0$ and $H = 0.5$ are most straightforward to see from *Equation 5*. When $H = 0$:

$$q(z_{1n}) \propto p(x_n|z_1)q(z_{1,n-1}),$$
$$q(z_{2n}) \propto p(x_n|z_2)q(z_{2,n-1}),$$

and

$$L_n = \log(p(x_n|z_1)/p(x_n|z_2)) + \log(q(z_{1,n-1})/q(z_{2,n-1})) = LLR_n + L_{n-1},$$

which is perfect integration of the log likelihood ratios. When $H = 0.5$:

$$q(z_{1n}) \propto p(x_n|z_1)[0.5q(z_{1,n-1}) + 0.5q(z_{2,n-1})],$$
$$q(z_{2n}) \propto p(x_n|z_2)[0.5q(z_{1,n-1}) + 0.5q(z_{2,n-1})],$$

and

$$L_n = \log(p(x_n|z_1)/p(x_n|z_2)) = LLR_n.$$

## Continuous-time model

Akin to the discrete-time model, the continuous-time version is based on the posterior probabilities of each option given all evidence collected until a given time point $t$. It has been shown previously that the non-normalized posterior probabilities of each of two states in a Markov jump process $dx(t)$, with average values $\pm\mu$ and noise magnitude $\sigma$, can be written as a system of stochastic differential equations (*Zakai, 1965*; *Liptser and Shiryaev, 1977*):

$$dq_1(t) = [-\lambda q_1(t) + \lambda q_2(t)]dt + q_1(t)\frac{\mu}{\sigma^2}dx(t),$$

$$dq_2(t) = [\lambda q_1(t) - \lambda q_2(t)]dt - q_2(t)\frac{\mu}{\sigma^2}dx(t). \tag{6}$$

We used this result to write the log-odds ratio signal as $L(t)$, seeking the derivative $dL(t) \equiv d\log(q_1(t)/q_2(t))$, by beginning with *Equation 6*, separating out the deterministic and stochastic components of the incoming evidence, and rewriting *Equation 6* in vector form:

$$d\mathbf{q}(t) = \left(\mathbf{L} + \frac{\mu}{\sigma^2}\mathbf{D}h(t)\right)\mathbf{q}(t)dt + \frac{\mu}{\sigma}\mathbf{D}\mathbf{q}(t)dW,$$

$$\mathbf{q}(t) \equiv \begin{pmatrix} q_1(t) & q_2(t) \end{pmatrix}^T \quad \mathbf{L} \equiv \begin{pmatrix} -\lambda & \lambda \\ \lambda & -\lambda \end{pmatrix} \quad \mathbf{D} \equiv \begin{pmatrix} 1 & 0 \\ 0 & -1 \end{pmatrix}. \tag{7}$$

.

Applying Itō's Lemma:

$$dL(t) = df(\mathbf{q}(t)) = \left[(\nabla f)^T\left(\mathbf{L} + \frac{\mu}{\sigma^2}\mathbf{D}h(t)\right)\mathbf{q}(t) + \frac{1}{2}\mathrm{Tr}\left[\left(\frac{\mu}{\sigma}\mathbf{D}\mathbf{q}(t)\right)^T(\nabla^2 f)\left(\frac{\mu}{\sigma}\mathbf{D}\mathbf{q}(t)\right)\right]\right]dt$$

$$+ \ldots(\nabla f)^T\left(\frac{\mu}{\sigma}\mathbf{D}\mathbf{q}(t)\right)dW,$$

$$f = \log(q_1(t)) - \log(q_2(t)) \quad \nabla f = \begin{pmatrix} 1/q_1(t) & -1/q_2(t) \end{pmatrix}^T$$

$$\nabla^2 f = \begin{pmatrix} -1\big/(q_1(t))^2 & 0 \\ 0 & 1\big/(q_2(t))^2 \end{pmatrix}.$$

$$\tag{8}$$

We now expand each component of *Equation 8*, beginning with those in the deterministic expression:

$$(\nabla f)^T \left( \mathbf{L} + \frac{\mu}{\sigma^2} \mathbf{D}h(t) \right) \mathbf{q}(t) =$$

$$\begin{pmatrix} 1/q_1(t) & -1/q_2(t) \end{pmatrix} \begin{bmatrix} -\lambda + h(t)\mu/\sigma^2 & \lambda \\ \lambda & -\lambda - h(t)\mu/\sigma^2 \end{bmatrix} \begin{pmatrix} q_1(t) & q_2(t) \end{pmatrix}^T = \quad (9)$$

$$1/q_1(t) \times \left( -\lambda + h(t)\mu/\sigma^2 \right) \times q_1(t) + 1/q_1(t) \times \lambda \times q_2(t) + \dots$$

$$-1/q_2(t) \times \lambda \times q_1(t) - 1/q_2(t) \times \left( -\lambda - h(t)\mu/\sigma^2 \right) \times q_2(t) =$$

$$2h(t)\mu/\sigma^2 + \lambda(q_2(t)/q_1(t) - q_1(t)/q_2(t)),$$

and

$$\frac{1}{2} \mathrm{Tr} \left[ \left( \frac{\mu}{\sigma} \mathbf{D}\mathbf{q}(t) \right)^T (\nabla^2 f) \left( \frac{\mu}{\sigma} \mathbf{D}\mathbf{q}(t) \right) \right] =$$

$$\frac{1}{2} \left( \frac{\mu}{\sigma} \right)^2 \begin{pmatrix} q_1(t) & -q_2(t) \end{pmatrix} \begin{pmatrix} -1 \big/ \left( q_1(t) \right)^2 & 0 \\ 0 & 1 \big/ (q_2(t))^2 \end{pmatrix} \begin{pmatrix} q_1(t) & -q_2(t) \end{pmatrix}^T = \quad (10)$$

$$\frac{1}{2} \left( \frac{\mu}{\sigma} \right)^2 \left[ -(q_1(t))^2 \big/ (q_1(t))^2 + (q_2(t))^2 \big/ (q_2(t))^2 \right] = 0.$$

Turning to the stochastic component:

$$(\nabla f)^T \left( \frac{\mu}{\sigma} \mathbf{D}\mathbf{q}(t) \right) = \frac{\mu}{\sigma} \begin{bmatrix} 1/q_1(t) & -1/q_2(t) \end{bmatrix} \begin{bmatrix} q_1(t) & 0 \\ 0 & -q_2(t) \end{bmatrix}$$

$$= \frac{\mu}{\sigma} \left[ q_1(t)/q_1(t) + q_2(t)/q_2(t) \right] = 2\frac{\mu}{\sigma}. \quad (11)$$

Substituting *Equations 9–11* into *Equation 8* yields:

$$dL(t) = \left[ -\lambda \times (q_1(t)/q_2(t) - q_2(t)/q_1(t)) + 2\frac{\mu}{\sigma^2} h(t) \right] dt + 2\frac{\mu}{\sigma} dW. \quad (12)$$

Letting $A = 2\mu/\sigma^2$, and using the hyperbolic sine function, we have $dL(t) = [-2\lambda \sinh L(t) + Ah(t)]dt + A\sigma dW$, which can be rewritten as *Equation 4* using $dx(t) = h(t)dt + \sigma dW$. Simulations in *Figure 1—figure supplement 1* show examples of time-evolution of the belief variable by approximating *Equation 4* with the Euler-Maruyama method.

## First-order approximations

We made first-order Taylor approximations of the deterministic terms in each model (*Figure 2A–D*, *Equation 3*).

### Discrete-time model

For the discrete-time model, we based the approximations on the log prior odds in *Equation 2*: $\psi(L_{n-1}, H) \approx \psi(L'_{n-1}, H) + \partial/\partial L_{n-1} \psi(L'_{n-1}, H)(L_{n-1} - L'_{n-1})$, where $L_{n-1}'$ is the value of the previous belief around which the approximation was made, and

$$\frac{\partial}{\partial L} \psi(L', H) = 1 - \frac{\exp(-L')}{\exp(-L') + \frac{1-H}{H}} - \frac{\exp(L')}{\exp(L') + \frac{1-H}{H}}$$

$$= 1 - \frac{1}{1 + \exp(L')\frac{1-H}{H}} - \frac{1}{1 + \exp(-L')\frac{1-H}{H}}.$$

Writing this equation with leak rate $K$ and bias $\theta$ as in **Equation 3**, $K \equiv 1 - \partial/\partial L \psi(L', H)$ and $\theta \equiv \psi(L', H) - \partial/\partial L \psi(L', H) L'$.

When previous beliefs are weak; that is, $L' = 0$,

$$\frac{\partial}{\partial L} \psi(L', H) = 1 - \frac{1}{1 + \frac{1-H}{H}} - \frac{1}{1 + \frac{1-H}{H}} = 1 - 2H,$$

and

$$\psi(L', H) = L' + \log\left[\exp(-L') + \frac{1-H}{H}\right] - \log\left[\exp(L') + \frac{1-H}{H}\right]$$

$$= \log\left[1 + \frac{1-H}{H}\right] - \log\left[1 + \frac{1-H}{H}\right] = 0.$$

Expressing the approximation in terms of **Equation 3**, leak rate $K = 2H$ and bias $\theta = 0$, as in **Equation 3a**.

When previous beliefs are strongly in favor of the first alternative; that is, $L_{n-1} \to \infty$,

$$\lim_{L' \to \infty} \frac{\partial}{\partial L} \psi(L', H) = 1 - 0 - \frac{1}{1 + 0 \times \frac{1-H}{H}} = 0,$$

so the leak rate $K = 1$, and the bias is determined entirely by the value of the log-prior odds evaluated as:

$$\lim_{L' \to \infty} \left[\psi(L', H)\right] = L' + \log\left[\frac{1-H}{H}\right] - \log\left[\exp(L')\right] = \log\left(\frac{1-H}{H}\right),$$

as in **Equation 3b**.

Similarly, when previous beliefs are strongly in favor of the second alternative; i.e., $L_{n-1} \to -\infty$,

$$\lim_{L' \to -\infty} \frac{\partial}{\partial L} \psi(L', H) = 1 - \frac{1}{1 + 0 \times \frac{1-H}{H}} - 0 = 0,$$

so the leak rate $K = 1$ here as well, and the bias is determined entirely by the value of the log-prior odds evaluated as:

$$\lim_{L' \to -\infty} \left[\psi(L', H)\right] = L' + \log\left[\exp(-L')\right] - \log\left[\frac{1-H}{H}\right] = -\log\left(\frac{1-H}{H}\right),$$

as in **Equation 3c**.

## Continuous-time model

We approximated continuous-time model in **Equation 4** by taking the first-order Taylor approximation of the deterministic term, which we write here as $g(L) \equiv -2\lambda\sinh(L)$. Specifically, $g(L) \approx kL + b$, where $k$ represents the time-varying leak rate as in the discrete-time model (**Figure 2B**), and $b$ represents a bias in the derivative of the belief variable (**Equation 4**; **Figure 2D**) that, along with the changing leak rate, effects a stabilizing boundary like in the discrete-time version. We calculated $k$ as the slope of $g(L)$, which is given as $2\lambda \times d/dL \sinh(L'(t)) = 2\lambda\cosh(L'(t))$, where $L'$ is the belief state around which the approximation was made. We computed bias as: $g(L', \lambda) - \frac{\partial}{\partial L} g(L', \lambda) \times L' = -2\lambda\sinh(L') + 2\lambda L'\cosh(L')$.

Analogous to the discrete-time case, when $L' = 0$ (certainty is low and beliefs are weak), $k = 2\lambda\cosh(0) = 2\lambda$, $b = -2\lambda\sinh(0) + 2\lambda \times 0 \times \cosh(0) = 0$, and the approximation is linear, resulting in an Ornstein-Uhlenbeck process during periods of stability in the data. However, as $L' \to \pm\infty$, $k \to \infty$, analogous to the leak rate approaching one in the discrete-time case, and

$$\lim_{L' \to \pm\infty} (b) = \lim_{L' \to \pm\infty} \left(-2\lambda\sinh(L') + 2\lambda L'\cosh(L')\right) =$$
$$\lim_{L' \to \pm\infty} \left[2\lambda L'\left(-\sinh(L')/L' + \cosh(L')\right)\right] = \lim_{L' \to \pm\infty} \left[2\lambda L'\cosh(L')\right] \to \pm\infty.$$

Whereas discrete-time approximations of the model give log-priors that are qualitatively similar to *Equation 2* for strong beliefs (*Figure 1C*), this regime in general is not as well approximated as a linear-Gaussian process, with steady-state solutions over stable periods that are shifted, extreme-value distributions (*Figure 1—figure supplement 1*, panel D). These distributions can be approximated by first solving for the general steady-state probability distribution of $L$, as follows. Beginning with the corresponding Fokker-Planck equation and letting p($L$, $t$) denote the time-dependent probability distribution of $L$ and $\gamma = Ah(t)$, the average of the sensory evidence during the stable period, we want the solution to p($L$, $t$) such that: $\frac{\partial}{\partial t} p(L, t) = -\frac{\partial}{\partial L}(-2\lambda\sinh(L) + \gamma)p(L, t) + \sqrt{\gamma^2}\frac{\partial^2}{\partial L^2}p(L, t) = 0$.

Therefore, $\frac{\partial}{\partial L}(-2\lambda\sinh(L) + \gamma)p(L, t) = \sqrt{\gamma^2}\frac{\partial^2}{\partial L^2}p(L, t)$, which we solved as $p(L) = C_0\exp\left(\int_{L_a}^{L}\frac{-2\lambda\sinh(L) + \gamma}{\sqrt{\gamma^2}}dL\right)$, where $C_0$ is a normalizing constant and $L_a$ is a reflecting boundary condition. So $p(L) = C_0\exp\left((-2\lambda\cosh(L) + \gamma L + 2\lambda\cosh(L_a) - \gamma L_a)/\sqrt{\gamma^2}\right)$ and letting $C$ be another normalizing constant that absorbs the constant terms inside the exponential:

$$p(L) = C\exp\left((-2\lambda\cosh(L) + \gamma L)\Big/\sqrt{\gamma^2}\right). \tag{13}$$

In the high-certainty regime (the expected value of the belief variable is very positive or negative, either because of strong sensory evidence or a very low hazard rate, or both), this expression can be well approximated as

$$p(L) \approx C\exp((-\lambda\exp(L) + \gamma L)/\gamma) \text{ when } L(t) \gg 0, \tag{14a}$$

$$p(L) \approx C\exp((-\lambda\exp(-L) + \gamma L)/-\gamma) \text{ when } L(t) \ll 0. \tag{14b}$$

*Figure 1—figure supplement 1*, panel B shows an example of this approximation along with simulations. *Equation 14* can be rewritten as extreme value distributions with location parameters $\pm\log(\gamma/\lambda)$ with the sign depending on the sign of the sensory evidence, and scale parameter = 1. For example, taking *Equation 14a*,

$$C\exp((-\lambda\exp(L) + \gamma L)/\gamma) = C\exp\left(-\frac{\lambda}{\gamma}\exp(L) + L\right) =$$

$$C\exp(-\exp(L - \log(\gamma/\lambda)) + L - \log(\gamma/\lambda) + \log(\gamma/\lambda)) =$$

$$C'\exp(-\exp(L - \log(\gamma/\lambda)) + L - \log(\gamma/\lambda)),$$

where $C' = C\gamma/\lambda$.

## Tasks

48 subjects (29 female, 19 male; age range = 19–45 years) participated in the triangles task, and 13 subjects (7 female, 6 male; age range = 19–38 years) participated in the dots-reversal task after providing informed consent. Human subject protocols were approved by the University of Pennsylvania Internal Review Board. Both tasks were performed on an iMac with a 27″ (68.5 cm) screen.

### Triangles task

Triangles were separated by 16 cm and represented the centers of a pair of two-dimensional Gaussian distributions. On each trial, one triangle was chosen as the true source of the generated star, and that source's associated two-dimensional distribution was sampled to determine the position of the star. Distributions were directly represented on the screen by scaling the color axis by screen position between blue and green according to the probability that a star would be generated in the given position (*Figure 4A,B*). Because the triangles were separated along the horizontal axis, only this dimension was relevant for determining which triangle represented the true source on that trial. For each trial, the star blinked on and off for ~1.5 s before the subject could choose the inferred source of that star, to minimize fast guesses. For each session, one of three different variances of the pair of two-dimensional distributions was randomly chosen without replacement (the ratio of the standard deviation of the generative

process to the distance between the triangles was 0.24, 0.33, or 0.41). These three conditions corresponded to mean values of log-likelihood ratios of 9, 4.5, or 3.33, respectively, of generated stars.

Each subject performed two 1000-trial blocks per session in 1–4 total sessions. Each block used a hazard rate that governed the rate of switching between the two sources (triangles) and was chosen randomly from a set of seven possible values (0.05, 0.1, 0.3, 0.5, 0.7, 0.9, 0.95). Hazard rates were chosen without replacement within sessions to ensure a change across blocks. In a subset of sessions, each block began with 400 trials in which subjects received feedback on the correct choice, followed by another 400 trials without feedback and ending with 200 feedback-based trials so that changes in hazard rate did not coincide with the onset of feedback.

Before each session, subjects were instructed that the triangles generated stars into overlapping neighborhoods and shown representations of the spatial distributions. They were then instructed that triangles would 'take turns' generating stars, with switches in the turn-taking that would occasionally be fast or slow. After receiving these instructions, subjects were shown an animation illustrating the generative process (i.e., a sample sequence of trials).

Subjects were paid a minimum $8 per session and an additional amount based on performance: at the beginning of each session, the subject had $5, and over the trials was penalized by 20 cents for each incorrect choice and rewarded with either 1 or 2 cents for correct choices. Total cash reward was continuously updated on feedback trials but not on non-feedback trials so subjects could not infer the previous correct choice. On average, subjects received a total additional cash reward of $8 (range $0–27).

## Dots-reversal task

This task was based on decisions about the coherent direction of motion of a set of stochastic dots (density = 70 dots/deg$^2$/s) presented in a 10°-diameter circular aperture at the center of the computer screen with three interleaved frames of motion. Each trial involved a stimulus 5–10 s in duration, determined as min(10, 5 + $\tau$), where $\tau$ was an exponentially distributed random variable, making trial terminations unpredictable within the given time frame. Within each trial, the direction of movement alternated between leftward and rightward at an average hazard rate of either 0.1 Hz or 2 Hz. Subjects participated in two sessions each, with 200 trials per session and a hazard rate that was constant throughout the session. The order of the two hazard rate sessions was chosen at random.

Each session began with 20 practice trials in which coherence was set to 60–85%, relatively high values that were considered easy for all subjects. For the remaining 180 trials, coherence was randomly chosen as either this 'high coherence' value (25% of trials) or a 'low-coherence' value (75% of trials). The value used for 'low coherence' was determined separately for each subject (mean ± SEM = 14.85 ± 1.53%, range 6–38%), corresponding to the coherence for which the participant could correctly decide the direction of a 500-ms long stable stimulus (i.e., no direction changes) 65% of the time. We assessed this threshold using a modified version of the adaptive QUEST procedure in which coherence was set to the mean of the threshold probability distribution (*Watson and Pelli, 1983*; *King-Smith et al., 1994*). We ran the QUEST procedure before each session and did not end the procedure until stable performance was achieved, defined as an estimated threshold log-likelihood ≥ −2.5 under the assumption that unstable performance would shift the corresponding threshold and keep log-likelihoods lower (direct estimates of threshold across consecutive 20-trial blocks verified this assumption). We also ensured that the measured threshold from this procedure was consistent across sessions for individual subjects. We based 'high' coherence on the low-coherence threshold: subjects with thresholds >15% received 85% high coherence, those with thresholds between 7–15% received 80% high coherence, and one subject with a remarkably low (~4–5%) threshold received 60% high coherence. All stimuli in this range are fairly easy to judge.

Before the session, subjects were instructed to do their best to follow motion direction throughout a given trial and indicate the direction they believed dots were moving in at the 'very end' of the trial. Subjects received feedback after each trial on the correct answer but were not given additional monetary reward for performance as in the triangles task.

## Model fitting

All models were fit to choice data using Matlab's optimization toolbox by minimizing the cross-entropy error function (*Bishop, 2006*):

$$e = -\sum_n (1-\rho_n)\log(1-\widehat{\rho}_n) + \rho_n \log(\widehat{\rho}_n), \tag{15}$$

where $\rho_n$ is a binary variable indicating which alternative was chosen on trial $n$ (arbitrarily defined as 0 for the right and 1 for the left) and $\widehat{\rho}_n$ is the choice probability predicted by the given model. All models assumed the choices were based on the sign of the subjective log-odds.

## Triangles task

We fit choices from the triangles task for each block of trials and each session by computing estimated $L_n$ on each iteration of the fitting algorithm using *Equation 1*; the current best estimate of subjective hazard rate for that iteration and block of trials ($\widehat{H}_b$); and a gain term on star position ($\beta$) that was applied across blocks but was specific to each session, which accounted for a subjective estimate of the generative variance (*Gold and Shadlen, 2001*): $L_n = \psi(L_{n-1}, \widehat{H}_b) + \beta x_n$. Unless otherwise indicated, all fits were made to trials without any feedback on the correct answer to confirm that learned hazard rates were used during this period. We omitted from further analyses the first 200 trials of each block for sessions in which no trial-by-trial feedback was given at all, allowing for stabilization of learned, subjective estimates of the current generative $H$ and resulting in a total of 800–6400 trials per subject for analysis. We assumed choices were based on the subjective log-odds and Gaussian noise: $\rho_n = \text{sign}(L_n + \zeta_n)$ where $\varsigma_n$ is the Gaussian noise variable. We also assumed a fixed magnitude of noise $\upsilon$ across sessions but specific to each subject. Model fits thus computed the cumulative probability of choosing the left triangle for subject $i$, in session $s$, block $b$, and trial $n$ as:

$$\widehat{p}_{i,s,b,n} = \frac{1}{2} + \frac{1}{2}\text{erf}\left(L_{i,s,b,n} \middle/ \sqrt{2}\upsilon_i\right), \quad L_{i,s,b,n} = \psi\left(L_{i,s,b,n-1}, \widehat{H}_{ib}\right) + \beta x_n. \tag{16}$$

Thus, this model had 4–16 free parameters for each subject, depending on the number of sessions. Because of the large number of parameters for some subjects, we used a maximum a posteriori (MAP) fitting procedure that placed conjugate priors on each parameter that were fixed across subjects and conditions, subtracting off log($p$[*parameter estimate*|*conjugate prior distribution of parameters*]) from the error term in *Equation 15*.

## Dots-reversal task

We fit the continuous-time model to data from the dots-reversal task using *Equation 1*, which is the most efficient discrete-time approximation of *Equation 4* and faster than the Euler-Maruyama approximation with small time-steps. However, unlike the triangles task, here all choices were made after observing an entire sequence of stimuli over the trial. We thus fit the model using *Equation 1* with $n$ indexing trial rather than time and $m$ indexing time-step within a trial, $i$ indexing subject and $s$ indexing session, as follows:

$$L_{ismn} = \psi\left(L_{is,m-1,n}, \widehat{H}_{is}\right) + k_i C_{ismn} + \sqrt{2k_i\langle|C_{isn}|\rangle}\eta_{ismn}, \tag{17}$$

where $k_i$ is a gain term for that subject on signed coherence $C_{ismn}$ (negative for rightward motion, positive for leftward motion; because coherence was determined probabilistically, its magnitude could vary step-by-step within a trial); $\langle|C_{isn}|\rangle$ indicates the expected coherence magnitude for that trial, determined experimentally as described above; $\eta_{ismn}$ is zero-mean, unit variance Gaussian noise; and $\widehat{H}_{is}$ is subjective hazard rate specific to each session. To convert to a continuous-time subjective hazard rate $\lambda$, we multiplied $\widehat{H}_{is}$ by the monitor refresh rate (60 Hz), which determines the time-steps between dot drawings.

The last two terms of *Equation 17* represent the subjective *LLR*, which we assume reflects a coherence-dependent signal (the second-to-last term) plus internal (neural), signal-dependent noise (the last term). Because we could not directly observe this noisy quantity, we fit the model by numerically deriving the time-evolution of the log-odds probability distribution over each trial and each step of the fitting iteration. Specifically, we determined the probability of each log-odds ($L_{isomn}$) at each time step by marginalizing over the probability of each log-odds from the previous time-step ($L_{isj, m-1, n}$):

$$p(L_{isomn}) = \int_{-\infty}^{\infty} \mathrm{N}\left(L_o \Big| \psi\left(L_j, \widehat{H}_{is}\right) + k_i C_{ismn}, \sqrt{2 k_i \langle |C_{isn}| \rangle}\right) p(L_{isj,m-1,n}) \, \mathrm{d}L_j, \tag{18}$$

where $\mathrm{N}(x|\mu,\sigma)$ denotes a Gaussian probability distribution function of variable $x$ with mean $\mu$ and standard deviation $\sigma$ and is the conditional probability distribution of the log-odds on each time-step given the coherence, model parameters, and log-odds on the previous time-step. Probability distributions were initialized on each trial as a Dirac delta function. For the model fits, we discretized the log-odds space between $-10$ and $10$ over steps of 0.4 log-odds, which we determined via simulations to be a sufficient range and resolution for accurate parameter estimates. Estimated choice probabilities were then computed as the cumulative probability of choosing the leftward direction: $\widehat{p}_{isn} = \int_0^{\infty} p(L_{isMn}) \mathrm{d}L_i$. Model fits were thus based on three parameters per subject, a single gain term $k_i$ and two hazard rate terms $\widehat{H}_{is}$, one for each session.

## Sub-optimal model #1: block-independent model

For both tasks, we fit to choice data a version of the normative model that ignored the block-wise changes in objective hazard rate. To ensure a fair model comparison with the block-dependent normative model, we constructed the null model using the same number of hazard-rate parameters but randomly shuffled across blocks. Specifically, for the triangles task the main model equation (*Equation 16*) was written as $L_{isbn} = \psi(L_{isb,n-1}, \widehat{H}_{is,b(n)}) + \beta_{is} x_{isbn}$, where $b(n)$ denotes the hazard-specific block to which this trial was randomly assigned. Randomization was performed before each fit, and this shuffle-and-fit procedure was repeated 50 times to generate a distribution of model log-likelihoods and BIC values against which the normative model could be compared across subjects. For the dots-reversal task, the main model equations (*Equations 17, 18*) were written as $L_{mn} = \psi(L_{m-1,n}, \widehat{H}(s_n)) + k C_{mn} + \sqrt{2k\langle |C_n| \rangle} \eta_{mn}$, where $s$ was a vector indicating the randomized session, with the shuffle-and-fit procedure repeated 20 times per subject.

## Sub-optimal model #2: block-dependent leaky accumulator

For both tasks, we also fit an alternative model based on the linear approximation in *Equation 3a*. For the triangles task this model was written as $L_{isbn} = (1 - K_{isb}) L_{isb,n-1} + \beta_{is} x_{isbn}$, where $K$ was the leak specific to subject $i$ in session $s$ and block $b$, and the other parameters were as described above. For the dots-reversal task we fit the same model, only here treating noise as a latent variable like we did for fitting the normative model and deriving the time-evolution of the log-odds probability distribution over each trial and each step of the fitting iteration: $L_{ismn} = (1 - K_{is}) L_{ism-1,n} + k_i C_{ismn} + \sqrt{2k_i \langle |C_{isn}| \rangle} \eta_{ismn}$. Unlike the normative model, these fits were greatly simplified by an analytic solution for the stationary standard derivation of the log-odds, given the current coherence and model parameters: $\sigma_{isn} = \sqrt{k_i \langle |C_{isn}| \rangle / K_{is}}$. This quantity is the standard deviation for the stationary probability distribution of the discrete-time analogue of an Ornstein-Uhlenbeck process in which coherence is perfectly stable over a trial. Thus, for each step of the fitting procedure, we solved for the average log-odds at the end of the trial by: (1) running the deterministic portion of the leaky-accumulation over the time-dependent coherence for the trial, and (2) writing the probability distribution of the log-odds at the end of the trial as a Gaussian with the final mean determined in step 1 and a standard deviation as described above, giving $\widehat{p}_{i,s,n} = \frac{1}{2} + \frac{1}{2} \mathrm{erf}(L_{i,s,M,n} / \sqrt{2k_i \langle |C_{isn}| \rangle / K_{is}})$, where $M$ indicates the final sample of trial $n$.

## Sub-optimal model #3: perfect accumulation with block-dependent stabilizing boundaries

For both tasks, we also fit an alternative model assuming perfect integration to a stabilizing boundary as in *Equation 2* with the boundary in *Equation 3b,c* as a free parameter; that is, rewriting *Equation 2* as:

$$\psi(L_{n-1}, H) = \begin{cases} L_{n-1} & -\theta < L_{n-1} < \theta \\ \theta & L_{n-1} \geq \theta \\ -\theta & L_{n-1} \leq -\theta \end{cases}, \tag{19}$$

$$\theta = \log\left(\frac{1-H}{H}\right).$$

For the triangles task, these fits were made as for the normative model, only using the re-written expression in *Equation 19* for log-prior odds above; that is, $L_{isbn} = \psi(H_{isb}, L_{isb,n-1}) + \beta_{is}x_{isbn}$. For the dots-reversal task, we similarly made fits as for the normative model, substituting the expression for $\psi$ in *Equation 19* into *Equations 17, 18*.

## Analysis details

Choice data shown in *Figure 5A* were fit by a two-parameter logistic function: $\hat{\rho}_n = 1/[1 + \exp((LLR_n - \phi)/\beta)]$, where $\phi$ represents the *LLR* for which subjects have a 50% chance of choosing either side, and $\beta$ is the slope of the function around that point. Average differences in parameter $\phi$ by hazard-rate condition are reflected in the horizontal shift of the psychometric function, representing a bias towards (leftward shift) or away from (rightward shift) repeating the same choice. Like for the fits of the main normative model, here the first 200–400 trials of each block were excluded to allow for a period of learning. We report the correlation between $\phi$ and the prediction from the asymptotic approximation of the fit normative model: $\log((1 - H_{subj})/H_{subj})$.

The subjective mappings of $\psi_n$ to $L_{n-1}$ in *Figure 6* were estimated as a non-parametric function, fit to choice data based on $L_n = \hat{\psi}(L_{n-1}) + LLR_n$ where here the *LLR* was based on the generative variance used for the task. We expressed $\psi_n$ as an interpolated function of $L_{n-1}$, with values spread evenly between −10 and 10 in steps of one log-odds ratio and interpolation performed with cubic splines. We fit the mapping with the same objective function as for the parametric models (*Equation 10*), using the entire data set within a given block to estimate the $\psi$ that best fit choice data. We used Tikhonov regularization of the derivative (for smoothness) using the added penalty term: $\gamma \sum_i (\Delta\psi_i)^2$, where $i$ indexes the value of $L$ for which $\psi$ was estimated and $\gamma = 1/20$ was determined through ad hoc methods.

Standard errors and statistical tests on performance measured as a function of viewing duration following the final change-point on the dots-reversal task were based on bootstrapped samples of the behavioral data or model predictions (*Figure 8A–F*, *Figure 9H–K*). Viewing duration bins were 0–200, 200–500, 500–1000, 1000–1500 and 1500–3000 ms. These analyses only included trials for which there was at least one change-point and the duration of the second-to-last direction was at least 300 ms to avoid immediately sequential change-points. The mean ± SEM durations of the second-to-last direction for trials used in this analysis were 2945 ± 55 ms for 0.1 Hz and 789 ± 13 ms for 2 Hz. At these durations, discrimination accuracy was likely to be at nearly asymptotic levels at the time of the final change-point. For a given duration bin specific to coherence and condition (indexed jointly as $k$), a single bootstrapped sample $m$ of performance was calculated as $\bar{\rho}_{km} = (\sum_n \bar{\rho}_{kmn})/N_{km}$, where $m$ indexes subject, generated as a random integer between zero and 14; $n$ indexes trial; and $N_{km}$ is the total number of trials within the duration bin for that subject and trial type (e.g., 0.1 Hz, low coherence). Means and standard errors were calculated as means and standard deviations of all bootstrapped samples (1000 samples per comparison). Statistical tests between condition-by-coherence trials $i$ and $j$ were based on paired differences between the same bootstrapped samples. The probability that $\bar{\rho}_i > \bar{\rho}_j$ was calculated as $\frac{1}{M}\sum_m (\bar{\rho}_{im} - \bar{\rho}_{jm} > 0)$, (where $M$ was the total number of bootstrapped samples for that comparison), likewise for $\bar{\rho}_i < \bar{\rho}_j$, and statistical significance indicated when one of these was less than the desired confidence level (0.05).

Fit leaks for the dots-reversal task (*Figure 9*) used the same algorithm as in the leaky-accumulator model described above but with separate leak terms for both hazard rate and low vs high coherence. Specifically, for subject $i$, session $s$, time-step $m$, trial $n$, and coherence level $c$, we defined the leak $L_{ismn} = (1 - K_{isc})L_{ism-1,n} + k_i C_{ismn} + \sqrt{2k_i\langle|C_{isn}|\rangle}\eta_{ismn}$. The noise was based on the stationary standard derivation of the log-odds, given the current coherence and model parameters: $\sigma_{isn} = \sqrt{k_i\langle|C_{isn}|\rangle/K_{isc}}$. We then fit the choice data to the predicted probability of a given choice for each trial, $\hat{p}_{i,s,n} = \frac{1}{2} + \frac{1}{2}\mathrm{erf}(L_{i,s,M,n}/\sqrt{2k_i\langle|C_{isn}|\rangle}/K_{isc})$, where $M$ indicates the final sample of trial $n$. We used a beta distribution prior on leak rate, letting the maximized model log probability for each subject be based on the sum of log likelihoods and this log prior probability, similar to what we did for the triangles task. We were interested in the dependence of leak on coherence level and comparing these dependencies between the choice data and model predictions. To control for overall level of leak by session (and subject), we computed the dependence as a normalized quantity: $d_{is} = (K_{is,high} - K_{is,low})/(K_{is,high} + K_{is,low})$ where the 'high' and 'low' subscripts denote leaks for high and low coherences. The comparison

between the data and normative model prediction of the dependence on hazard rate (session) used an analogous normalized measure: $d_{is} = (K_{is,fast} - K_{is,slow})/(K_{is,fast} + K_{is,slow})$ where 'fast' and 'slow' indicate the 2 Hz and 0.1 Hz sessions respectively. Statistics reported were based on these difference measures.

## Acknowledgements

We thank Gabriel Kroch and Timothy Kim for help with data collection and Joe McGuire, Yin Li, Matt Nassar, and Bob Wilson for comments.

## Additional information

### Funding

| Funder | Grant reference | Author |
| --- | --- | --- |
| National Institutes of Health (NIH) | OppNet Grant R01 MH098899 | Christopher M Glaze, Joseph W Kable, Joshua I Gold |

The funder had no role in study design, data collection and interpretation, or the decision to submit the work for publication.

### Author contributions

CMG, Conception and design, Acquisition of data, Analysis and interpretation of data, Drafting or revising the article; JWK, JIG, Conception and design, Analysis and interpretation of data, Drafting or revising the article

### Ethics

Human subjects: Informed consent, and consent to publish, was obtained from each subject prior to each experiment. Human subject protocols were approved by the University of Pennsylvania Internal Review Board.

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
