## [Decision Letter]

Thank you for submitting your work entitled “Normative evidence accumulation in unpredictable environments” for peer review at *eLife*. Your submission has been favorably evaluated by Timothy Behrens (Senior and Reviewing Editor) and three peer reviewers.

The reviewers have discussed the reviews with one another and the editor has drafted this decision to help you prepare a revised submission.

The study of how the brain integrates a history of evidence into a single expectation has been a remarkably profitable avenue of research in forming quantitative understandings of neural mechanisms over recent years and decades. However, two largely distinct literatures have tackled this problem from very different perspectives. One set of researchers have considered the mathematics and neurophysiology of integrating continuous streams of evidence using approaches such as the drift diffusion model and the sequential probability ratio test. Another set of researchers have considered the integration of discrete events or trials using Bayesian models that can, for example, optimally determine when to integrate evidence together, and when to separate new evidence because the world has changed.

The current paper presents an important step towards a unification of these two literatures, by presenting an elegant mathematical analysis of the Bayesian change-point models to demonstrate that they can be viewed as modified sequential probability ratio tests, and are therefore directly applicable to the large set of researchers who are interested in continuous evidence integration. It makes new and interesting predictions about the stream integration case in situations when the evidence is non-stationary and it tests these predictions in behaviour. All three reviewers considered this to be a very interesting and potentially important step.

As you will see below, the reviewers did not have major criticisms of the model that is presented or data, which they broadly found to be convincing. Most important in terms of solidifying the conclusions, however, are Reviewer 1's and 3's similar points about the demonstration of the qualitative reason for the improvement of the current model (Reviewer 1) and the absence of a reasonable analysis of other (sub-optimal) models (Reviewer 3).

In the discussion, the reviewers and editor were also clear that the manuscript is really written for a technical audience. *eLife* readership is broad and the reviewers and editor would appreciate a reframing of the paper that makes the central points clearer to this broad audience.

One suggestion that emerged during the discussion was a clearer framing of the manuscript as follows:

a) The same Bayesian learning framework, that has been used in other contexts (work by Behrens, Adams, Kording and also Nassar and Gold) is here derived for evidence accumulation in perceptual decision-making tasks like RDM;

b) This normative framework can be described, in both discrete and continuous cases, as a leaky accumulator with non-absorbing bounds, with the leak rate and the height of the bounds being explicit functions of the environmental change rate;

c) Show that this is the case in their data (their paradigm with variable hazard rate is a good test even if atypical of RDM experiments);

d) Provide some justification why this model helps our understanding of perceptual decision-making (can the relationship with change point detection explain or offer insight into previously unexplained data?);

e) Provide some clear commentary on the relationship to previous models of change point detection (can the relationship with RDMs explain or offer insight into previously unexplained data?).

We urge you to consider this, or other possible reframings in which the strong message can be understood clearly without an understanding of the technical details.

Reviewer #1:

In this study, Glaze, Kable and Gold present a model of evidence integration over time in which the relative weighting given to new and past observations can be adjusted to reflect the hazard rate for change in the environment. They show that this model can be generalized to both discrete and continuous time cases and that it fits human performance better than a model without adjustable hazard rate, in two very different decision-making tasks – a random dot motion task with within-trial reversals, and a task in which evidence is accumulated over many discrete trials.

This is a good quality paper which I think will be of interest to people across the field of perceptual decision-making, since it presents a clear framework that is applicable in diverse tasks. It could be an influential and highly cited paper in the field.

If I were to make a case why the manuscript should be published in *eLife*, I would point out that the adaptation of the Bayesian framework to the random dot motion case does represent a major conceptual advance over the more typical approaches in that field (SPRT models with fixed leak rates or the drift diffusion model), and the model does better at explaining participants' performance than those more typical models. However, I would then suggest that the paper could be strengthened by exploring in more detail why the current model outperforms others (especially, the leaky accumulator with hazard rate as a free parameter by block). Is this because the leaky accumulator down weights past beliefs about the correct response without taking into account evidence strength? If so the reason for the current model's superiority is not really to do with estimating the hazard rate, although the text implies that it is. Furthermore, to what extent does the current model address burning questions in that field, such as how confidence judgements are made or when the evidence accumulation process should terminate?

Reviewer #2:

In this manuscript by Glaze et al., the authors present a normative model for evidence accumulation in a non-stationary environment in which the successive simples are drawn from one of two alternative distributions for an unknown and variable duration. The main part of the manuscript is to compare the performance of human subjects in two different behavioral tasks with the predictions of this normative model and a simpler alternative model based on leaky integration. The results show that human subjects differ significantly from the predictions of normative model, in that subjective estimate of the rate of change (hazard rate) is close to 0.5, suggesting that they tend to give insufficient weight to the history of evidence. Although the manuscript is quite technical in nature, it is written clearly, and the findings would be of high value to many researchers in the field. There are only a few, relatively minor, comments:

1) The overall conclusion of the authors is that the subjective estimates of hazard rate are biased, but this quantity is used in a normative way. However, this might be misleading. Namely, is it fair to refer to an algorithm as “normative” if the quantity used in this algorithm is biased? Does such an algorithm behave differently, for example, from a model that uses the accurate estimate of hazard rate in non-normative way? Unless the authors can clarify how these two different scenarios can be distinguished, how the word “normative” is used in this manuscript might need to be improved.

2) In the subsection “Psychophysics”, the authors should indicate for how many sessions, trial-by-trial feedback was provided in the beginning and end of each block.

3) What do green and blue colors in Figure 7 indicate?

Reviewer #3:

This is an interesting paper that provides a significant contribution through the clear derivation of a normative approach for accumulation of evidence under conditions where the evidence is not stationary. The derivation and the description of how changes in the rate of change of environments correspond to leakiness in accumulation was very appealing.

I found the experimental section convincing in terms of showing that humans do indeed try to estimate the rate of change of the environment, and that this estimate affects how they make decisions. But the further claim made in the paper that humans behave according to the normative model was harder to be convinced by – it seemed that other (suboptimal) models that use an estimate of the hazard rate might also be consistent with the data, but this wasn't much explored in the paper. Instead, the straw man such as a model where the estimated hazard rate is a single value fixed for all time was used, but it seemed rather a weak straw man.

In addition, much greater clarity in the exposition would be desirable.

Despite these concerns, the nice derivation of the normative results under changes in environments makes me see the paper positively.

---

## [Author Response]

*[…] As you will see below, the reviewers did not have major criticisms of the model that is presented or data, which they broadly found to be convincing. Most important in terms of solidifying the conclusions, however, are Reviewer 1's and 3's similar points about the demonstration of the qualitative reason for the improvement of the current model (Reviewer 1) and the absence of a reasonable analysis of other (sub-optimal) models (Reviewer 3)*.

Our original submission included direct comparisons of the normative model with two suboptimal models: 1) subjective hazard rate was randomly shuffled across conditions, meant to mimic choices under the assumption of a single subjective hazard rate, but with the same parameter structure; and 2) a leaky accumulator, which was inspired by both the approximation to the normative model in the weak-belief regime ([Disp-formula equ4] in the manuscript) as well as the many neural models that assume (linear) leaky integration. We now include a comparison to a third, suboptimal model, in this case assuming perfect integration to a stabilizing boundary (“bounded accumulation”), inspired by the approximation to the normative model in the strong-belief regime ([Disp-formula equ5 equ6]). We also now provide for each experimental task a consolidated discussion of how the fits compare between the normative model versus the leaky and bounded accumulator approximations and highlight key differences between models in Figures 3, 6 and 9.

*In the discussion, the reviewers and editor were also clear that the manuscript is really written for a technical audience.* eLife *readership is broad and the reviewers and editor would appreciate a reframing of the paper that makes the central points clearer to this broad audience*.

One suggestion that emerged during the discussion was a clearer framing of the manuscript as follows:

a) The same Bayesian learning framework, that has been used in other contexts (work by Behrens, Adams, Kording and also Nassar and Gold) is here derived for evidence accumulation in perceptual decision-making tasks like RDM;

*b) This normative framework can be described, in both discrete and continuous cases, as a leaky accumulator with non-absorbing bounds, with the leak rate and the height of the bounds being explicit functions of the environmental change rate*;

*c) Show that this is the case in their data (their paradigm with variable hazard rate is a good test even if atypical of RDM experiments)*;

*d) Provide some justification why this model helps our understanding of perceptual decision-making (can the relationship with change point detection explain or offer insight into previously unexplained data?)*;

*e) Provide some clear commentary on the relationship to previous models of change point detection (can the relationship with RDMs explain or offer insight into previously unexplained data?)*.

*We urge you to consider this, or other possible reframings in which the strong message can be understood clearly without an understanding of the technical details*.

We very much appreciate the thoughtful suggestions, which we have incorporated throughout the manuscript, including the Abstract, Introduction, Results, and Discussion sections.

Reviewer #1:

[…] However, I would then suggest that the paper could be strengthened by exploring in more detail why the current model outperforms others (especially, the leaky accumulator with hazard rate as a free parameter by block). Is this because the leaky accumulator down weights past beliefs about the correct response without taking into account evidence strength?

Yes, and this failure to account for evidence strength will lead to, for example, confidence that is allowed to grow to a much higher asymptote that hinders change detection when the leaky accumulator has been optimized to process weak evidence, as in [Disp-formula equ4]. In contrast, the normative model limits confidence at any given moment to the logarithmic terms in [Disp-formula equ5 equ6]. We now provide these intuitions, along with complementary limitations of the suboptimal model with perfect accumulation to a stabilizing boundary, in the new Figure 3 and associated text.

*If so the reason for the current model's superiority is not really to do with estimating the hazard rate, although the text implies that it is*.

We respectfully disagree: the value of the limit in confidence mentioned above depends entirely on estimated hazard rate as in [Disp-formula equ5 equ6]. In other words, the normative model accounts for evidence strength in a way that depends on hazard rate.

Furthermore, to what extent does the current model address burning questions in that field, such as how confidence judgements are made or when the evidence accumulation process should terminate?

We agree that these are very interesting issues that are closely related to evidence accumulation and therefore are pertinent to our model. We have an entire paragraph in the Discussion about how our model provides new insights into the speed-accuracy trade-off and decision termination. We thank the reviewer for pointing out the link to confidence judgments, which we now also include in that paragraph.

Reviewer #2:

*[…] There are only a few, relatively minor, comments*.

1) The overall conclusion of the authors is that the subjective estimates of hazard rate are biased, but this quantity is used in a normative way. However, this might be misleading. Namely, is it fair to refer to an algorithm as “normative” if the quantity used in this algorithm is biased?

We would argue that the algorithm is normative while the subjective parameter estimates are not. We now explain our perspective on this important point more clearly in the Discussion, as follows:

“[W]e showed that subjects could both learn a range of hazard rates and then use those learned rates in a normative manner to interpret sequences of evidence to make decisions. However, they tended to learn imperfectly, over-estimating low hazard rates and under-estimating high hazard rates. […] These different set points may have reflected certain prior expectations about the improbability of either perfect stability or excessive instability that could constrain performance when those conditions occur.”

*Does such an algorithm behave differently*, *for example, from a model that uses the accurate estimate of hazard rate in non-normative way?*

We agree that this is a very interesting question. We have provided several lines of evidence that support the idea that the subjects are using their subjective estimates of hazard rate in a normative fashion. First, we used choice data to directly estimate the mapping of subjective beliefs to priors (new Figure 6) and showed that subjects exhibited leaky/asymptotic integration in ways that closely matched predictions of the normative model using imperfect estimates of hazard rate. Second, their behavior exhibited the trade-off between change detection and steady-state signal identification predicted by the model, based on their subjective estimates of hazard (Figures 5 and 8). Third, their behavior also reflected the signal-strength-dependent biases predicted by the model, in a manner also dependent on their subjective estimates of hazard.

In addition, we now include in the manuscript estimated subjective hazard rates from both the leaky and bounded accumulator models for both tasks. These parameters also deviate from objective values, “further supporting the idea that the subjects were using imperfect estimates [of hazard rate] to make their decisions.”

We of course cannot fully rule out the possibility that there are some other models that can better explain our choice data using objective hazard rates. However, given the relatively few parameters per block in our model, plus its ability to capture many aspects of the behavioral data, we believe that we have presented a compelling case that, to a very large degree, our subjects’ behavior is consistent with a normative process using imperfect estimates of hazard rate.

*Unless the authors can clarify how these two different scenarios can be distinguished, how the word “normative” is used in this manuscript might need to be improved*.

We hope that we have addressed this important point sufficiently thoroughly and clearly in the revised manuscript.

*2) In the subsection “Psychophysics”, the authors should indicate for how many sessions, trial-by-trial feedback was provided in the beginning and end of each block*.

We apologize for the lack of detail, which we now provide.

*3) What do green and blue colors in*
Figure 7
*indicate?*

We apologize for the confusion and have now ensured that all figures have appropriate legends.

Reviewer #3:

*[…] I found the experimental section convincing in terms of showing that humans do indeed try to estimate the rate of change of the environment, and that this estimate affects how they make decisions. But the further claim made in the paper that humans behave according to the normative model was harder to be convinced by – it seemed that other (suboptimal) models that use an estimate of the hazard rate might also be consistent with the data, but this wasn't much explored in the paper. Instead, the straw man such as a model where the estimated hazard rate is a single value fixed for all time was used, but it seemed rather a weak straw man*.

We strongly agree for the need to compare normative model fits with other likely candidate models. We had originally focused on comparison with a leaky integrator that had the leak vary freely by blocks of trials, because this alternative model seemed to be the most viable form of information accumulation examined in prior studies that could solve these tasks. We apologize for any confusion about what forms of this model we used (it was not the case that we used the true straw man of a single fixed hazard rate for all conditions). We have clarified and expanded upon these issues substantially and include for each task model comparisons to both leaky integration (allowing the leak to vary by hazard-specific block) and bounded, perfect, accumulation (allowing the bound to vary by hazard-specific block). The models are described in detail in separate sub-sections of Methods.

*In addition, much greater clarity in the exposition would be desirable*.

We have implemented the suggested changes described in the summary comments, above, and hope that we have substantially clarified the exposition.